# Free-Lunch Color-Texture Disentanglement for Stylized Image Generation

**Jiang Qin[1,*], Alexandra Gomez-Villa[3,4,*], Senmao Li[2,*,‡],**
**Shiqi Yang[2,†], Yaxing Wang[2], Kai Wang[5,6,3,‡], Joost van de Weijer[3,4]**

[1]Harbin Institute of Technology, China [2]VCIP, CS, Nankai University, China
[3]Computer Vision Center, Spain [4]Universitat Autònoma de Barcelona, Spain
[5]Program of Computer Science, City University of Hong Kong (Dongguan), China
[6]City University of Hong Kong, HK SAR, China
https://deepffff.github.io/sadis.github.io

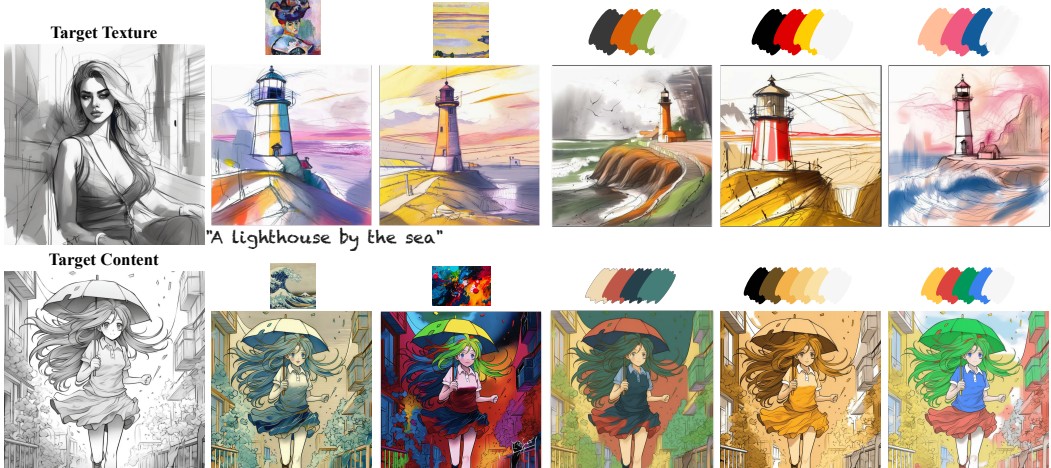

Figure 1: Stylized images generated by our *training-free* method, *SADis*. **(Up)** Example of texture (left) and color (above) conditioned text-to-image (T2I) generation. This approach offers creators enhanced color control, including the use of color palettes as shown in the last two columns. **(Down)** Example with conditioning on content image (left) and color (above), showing it extends to color-only stylized image generation.

## Abstract

Recent advances in Text-to-Image (T2I) diffusion models have transformed image generation, enabling significant progress in stylized generation using only a few style reference images. However, current diffusion-based methods struggle with *fine-grained* style customization due to challenges in controlling multiple style attributes, such as *color* and *texture*. This paper introduces the first tuning-free approach to achieve *free-lunch color-texture disentanglement* in stylized T2I generation, addressing the need for independently controlled style elements for the Disentangled Stylized Image Generation (*DisIG*) problem. Our approach leverages the *Image-Prompt Additivity* property in the CLIP image embedding space to develop techniques for separating and extracting Color-Texture Embeddings (*CTE*) from individual color and texture reference images. To ensure that the color palette of the generated image aligns closely with the color reference, we apply a whitening

---

[*]Equal contribution.
[†]Visiting researcher in Nankai University.
[‡]Corresponding authors: Senmao Li, Kai Wang

39th Conference on Neural Information Processing Systems (NeurIPS 2025).

and coloring transformation to enhance color consistency. Additionally, to prevent texture loss due to the signal-leak bias inherent in diffusion training, we introduce a noise term that preserves textural fidelity during the Regularized Whitening and Coloring Transformation (*RegWCT*). Through these methods, our Style Attributes Disentanglement approach (*SADis*) delivers a more precise and customizable solution for stylized image generation. Experiments on images from the WikiArt and StyleDrop datasets demonstrate that, both qualitatively and quantitatively, *SADis* surpasses state-of-the-art stylization methods in the *DisIG* task.

# 1   Introduction

Stylized Image Generation [11, 30, 36] aims to transfer a style from a reference image to a target image. This field has evolved through several technological paradigms, beginning with CNN-based feature manipulation [20, 65, 36], advancing to attention mechanisms [4, 75, 46], and further developing through GAN-based image translation [37, 83, 73]. A significant advancement came with multimodal CLIP guidance [19, 40, 51], which bridged the gap between visual and textual representations. This breakthrough enabled a novel approach where style transfer could be guided by textual descriptions rather than being limited to reference images [19, 41, 35]. The field underwent another transformation with the emergence of Text-to-Image (T2I) diffusion models [56, 59, 57], which demonstrated remarkable capabilities in personalized image generation [58, 18, 48, 7].

T2I diffusion models catalyzed new developments in stylized image generation [54, 67, 62] — a specialized paradigm that focuses on creating new images that incorporate specific visual characteristics from reference styles, rather than simply transferring style between existing images. Initially, many approaches [18, 2] relied on extensive fine-tuning using datasets of similarly styled images. However, this requirement proved impractical in real-world scenarios, where collecting cohesive style-specific datasets is often challenging. Addressing these limitations, recent research has focused on developing *tuning-free (free-lunch)* methods [76, 54, 38]. These approaches eliminate the need for costly retraining while maintaining efficient style integration capabilities. Despite their methodological differences, both tuning-based and tuning-free approaches share a common characteristic: they transfer entangled style representations during the image generation process, simultaneously incorporating both color and texture elements from the style images into the final output.

Although existing stylized image generation methods offer promising advancements, for content creators, these methods have critical limitations. A key challenge is the lack of granular control over style elements. Content creators often need to adopt specific aspects of a reference style while preserving elements of their original vision [13, 27]. Color palettes, in particular, play a crucial role in this process — they are often meticulously crafted to evoke specific emotions or maintain brand consistency. Creators may wish to preserve these while adopting textural[4] elements from other reference styles. However, current approaches force an all-or-nothing choice, where accepting a reference style means incorporating all its visual (texture and color) characteristics simultaneously, seriously limiting artistic freedom and practical applicability.

To address these limitations, we introduce the problem of *Disentangled Stylized Image Generation* (*DisIG*), which aims to decompose reference styles into independently controllable attributes like color, texture, and semantic content. This formulation enables artists to selectively transfer specific style elements while maintaining control over the content using text prompts. One potential workaround for *DisIG* using existing methods is to specify desired style attributes through text prompts. However, this approach requires extensive prompt engineering [47, 77, 72] expertise and considerable trial-and-error to achieve results that can be obtained more intuitively through image prompts. Even with significant effort in crafting precise textual descriptions, text prompts often fall short of capturing the nuanced characteristics of the target style. This limitation stems from the inherent expressiveness gap between text and visual information — compared with text prompts, image prompts inherently contain more fine-grained semantic information that is difficult to describe

---

[4]In this paper, we use the term *texture* to refer to the arrangement, repetition, and local patterns of visual elements such as shapes, edges, intensities, and gradients and excluding color aspects.

accurately in text, including style attributes such as color and texture. Such nuanced details contribute to improved generation quality (as shown in Fig. 2-2nd row.).

In this work, we propose what we believe to be the *first DisIG* method that enables independent control of color and texture through separate reference images, without any training. Such application and performance can be seen from Fig. 1. To achieve this, we first analyze the *additivity property* — a characteristic previously studied in textual spaces [50, 6, 33] — and demonstrate that it also holds true in the *image prompt* space. Then, we work with each style component separately: we isolate the *color representation* through feature subtraction between the original color image and its grayscale equivalent, while deriving the *texture representation* using a grayscale version of the texture reference image and SVD rescaling. Finally, to ensure accurate color matching while preserving textures, we introduce *RegWCT*, an enhanced whitening and coloring transform with noise regularization, which counteracts the *signal-leak bias* [16, 80] in diffusion models.

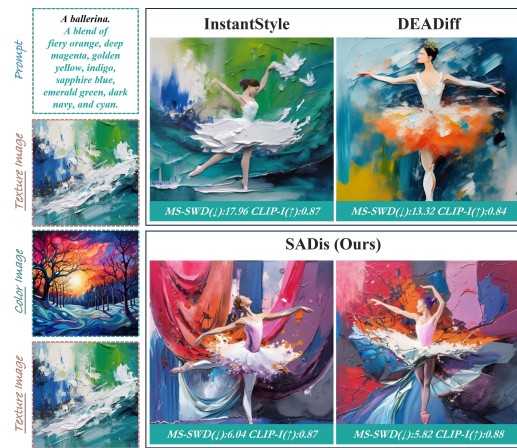

Figure 2: Compared to SOTA stylization methods, *SADis* enables more precise and fine-grained control of color and texture by leveraging style element images, which allows for more accurate specification of visual attributes[5].

In experiments, we use images from the WikiArt [64] and StyleDrop [62] datasets and evaluate our approach based on the SDXL model [53]. By comparing against several state-of-the-art stylization methods, *SADis* consistently outperforms these baselines over both qualitative and quantitative results, particularly in achieving accurate color and texture expression for the stylized T2I generation. In summary, our contributions are as follows:

- We introduce the *Disentangled Stylized Image Generation* (*DisIG*) problem, a critical challenge in real-world applications that enables richer semantic control in diffusion-based stylization.

- We are the *first* to identify and use the *image-prompt additivity* property in the CLIP image encoder, showing how image features can be decomposed and combined similarly to text embeddings.

- We present the *first* tuning-free (free-lunch) *style attribute disentanglement* method, *SADis*, emphasizing color and texture as key style attributes. Our color-texture extraction (*CTE*) and regularized whitening-coloring transformation (*RegWCT*) techniques enable effective disentangled color-texture stylization without additional tuning.

- Through thorough qualitative and quantitative evaluations in standard style transfer benchmarks, *SADis* consistently outperforms existing baselines in realistic generation under the *DisIG* scenario.

## 2  Related Work

Stylized image generation [30, 11, 37, 83], also related to classical style transfer, aims to generate images with the artistic style of a reference image. Early research focused on statistical feature manipulation to transfer artistic styles from reference images. Gatys et al. [20] pioneered this direction by using covariance matrices for style representation. Following works like AdaIN [36] improved efficiency by transferring feature statistics between style and content feature maps, while WCT [45] introduced whitening and coloring transformations to match covariance matrices. Recent architectural innovations have enhanced stylization capabilities.

A significant paradigm shift occurred with the introduction of CLIP [55], enabling connections between text and image representations in a shared embedding space. This advancement spawned methods like ClipStyler [41] to combine global and patch-level CLIP losses for image stylization.

---

[5]Here we use the prompt "A ballerina" as an example. The prompt used in InstantStyle and DEADiff is a detailed description of the color reference image generated by GPT-4o [1].

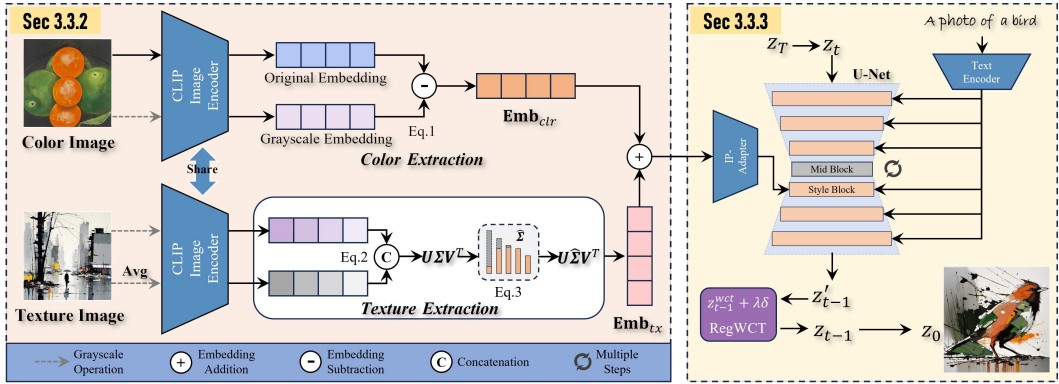

Figure 3: Our method, *SADis*, begins with color-texture extraction (*CTE*), leveraging the Image-Prompt Additivity property, which is verified in this paper for the *first* time. The color embedding $\mathbf{Emb}_{clr}$ is obtained by exploiting the Image-Prompt Additivity property, while the texture embedding $\mathbf{Emb}_{tx}$ is extracted via a singular value decomposition (SVD) operation. Afterwards, we incorporate these embeddings into the style cross-attention layer of the SDXL model. Subsequently, we refine the latent variable $z_{t-1}$ at each inference step with our proposed *RegWCT* transformation, aligning color palettes precisely while retaining essential texture details.

Nowadays, T2I Diffusion models [61, 29, 8] have emerged as the new state-of-the-art models for text-to-image generation. Then the existing T2I stylization methods [23, 60, 12, 81, 66, 22, 82, 69, 43] achieve stylized image generation via fine-tuning generative models on few style reference images. However, this process is time-consuming and struggles to generalize to real-world scenarios where gathering a suitable subset of shared-style images can be difficult. To address these limitations, interest in tuning-free (free-lunch) approaches for stylized image generation has grown [10, 15, 70, 71, 26, 24]. These methods introduce lightweight adapters that extract style information from reference images and inject it into the diffusion process via self-attention or cross-attention layers. Representative examples include IP-Adapter [76] and Style-Adapter [71], which employ a decoupled cross-attention mechanism that separates cross-attention layers for handling text and image features independently. DEADiff [54] introduces a different approach by focusing on extracting disentangled representations of content and style using paired datasets.

Although existing stylization methods show promise, they lack granular control over individual style elements - a crucial need for content creators who often want to selectively adopt specific aspects of reference styles. This introduces a challenge we term the *Disentangled Stylized Image Generation* (*DisIG*). Focusing on color and texture as the most significant style attributes, this paper introduces the *first* free-lunch approach to *color-texture disentanglement* for stylized image generation.

## 3 Method

### 3.1 Preliminaries

**T2I Diffusion Models.** We built on the SDXL [53] model, consisting of two primary components: an autoencoder and a diffusion model $\epsilon_\theta(z_t, t, \tau_\xi(\mathcal{P}))$, where $\epsilon_\theta$ is a UNet, conditioning a latent input $z_t$, a timestep $t \sim \mathrm{U}(1, T)$, and a text embedding $\tau_\xi(\mathcal{P})$. More specifically, text-guided diffusion models generate an image from the textual condition as $\mathcal{C}_{text} = \tau_\xi(\mathcal{P})$, where $\tau_\xi$ is the CLIP text encoder [55][6]. The cross-attention map is derived from $\epsilon_\theta(z_t, t, \mathcal{C}_{text})$. After predicting the noise, diffusion schedulers [63, 49] are used to predict the latent which we simplify as $z_{t-1} = \mathcal{G}(z_t, t, \mathcal{C}_{text})$.

**IP-Adapter.** Building on T2I diffusion models, the IP-Adapter [76] introduces additional controllability by conditioning the T2I model on a conditional image $\mathcal{I}_{ip}$. Practically, this involves leveraging a pre-trained T2I diffusion model and incorporating a cross-attention layer to the (projected) image condition following each text-prompt conditioning layer. The conditional image is encoded in the low-dimensional CLIP image embedding space [55] to capture high-level semantic information. By denoting the CLIP image encoder as $\tau_\phi$ and IP-Adapter projection as $\mathbf{IP}$, this process is adding a new image condition $\mathcal{C}_{img} = \mathbf{IP}(\tau_\phi(\mathcal{I}_{ip}))$ to the T2I model as $z_{t-1} = \mathcal{G}(z_t, t, \mathcal{C}_{text}, \mathcal{C}_{img})$.

---

[6]SDXL uses two text encoders and concatenate the embeddings.

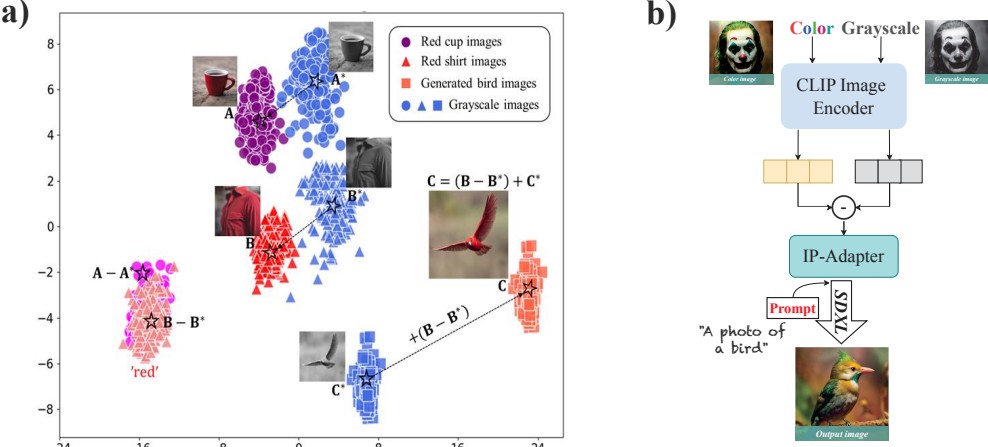

Figure 4: a) **Image-prompt additivity**: Image encoder latent space enables color extraction via grayscale subtraction. This color information can then be added to latent representations of other grey-scale images; b) By subtracting the grayscale image embeddings from the color image embeddings, we can effectively remove the texture information and isolate the color information.

## 3.2 Disentangled Stylized Image Generation

In a typical artistic workflow, creators draw inspiration from multiple sources to produce their final results. They might select a color palette from a sunset photograph, adopt textural patterns from a baroque painting, and incorporate structural elements from architectural blueprints—all while maintaining precise control over which elements they adopt from each reference. We term this ideal multi-target style transfer problem as Disentangled Stylized Image Generation (*DisIG*).

This multi-source approach to artistic creation (*DisIG*) contrast to current computational stylization methods [45, 9, 40], which typically extract and apply a bundled representation of "style" from a single reference image. To address the limitation, we first analyze how color and texture are entangled in existing stylization approaches. We focus specifically on these two attributes because they represent fundamental, measurable components of visual style that artists frequently manipulate independently.

**Problem Setup.** We formally define the *DisIG* task as follows: Given a text prompt $\mathcal{P}$, a color reference image $\mathcal{I}_{clr}$, and a texture reference image $\mathcal{I}_{tx}$, our goal is to generate an image that semantically aligns with $\mathcal{P}$ while independently adopting the color distribution of $\mathcal{I}_{clr}$ and the textural characteristics of $\mathcal{I}_{tx}$. The color reference $\mathcal{I}_{clr}$ can be either a standard RGB image or a discrete color palette. Unlike traditional style transfer that applies a holistic "style" from a single reference, our formulation enables precise, attribute-specific control by explicitly disentangling and separately transferring color and texture components.

## 3.3 Style Attributes Disentanglement

Here, we present our Style Attributes Disentanglement method (*SADis*). An overview is provided in Fig. 3. To successfully extract the color and texture information, we exploit the image-prompt additivity property as in Sec. 3.3.1. Next, we explain the color-texture extraction (*CTE*) in Sec. 3.3.2, and the noise-regularized whitening-coloring transformation (*RegWCT*) is detailed in Sec. 3.3.3.

### 3.3.1 Image-Prompt Additivity

To achieve disentangled stylization, we first analyze the *additivity property* of *image prompts*. This property is the main insight on which our color-texture disentanglement is based. It is illustrated using the CLIP image encoder, as adopted in the IP-Adapter [76]. Note that this property has been explored in text prompt spaces [50, 6]; here we show that it also holds in the image embedding space.

Consider the image embedding space as shown in Fig. 4-(a). Here we show a PCA plot of the embedding space for a distribution of images generated with three different text prompts. The embedding of a color image contains all information of the image (see e.g. points $A$ and $B$). If we compare this embedding with the embedding of its grey-scale version ($A'$, $B'$), the difference between the two embeddings is due to the removal of color information. We here hypothesize that

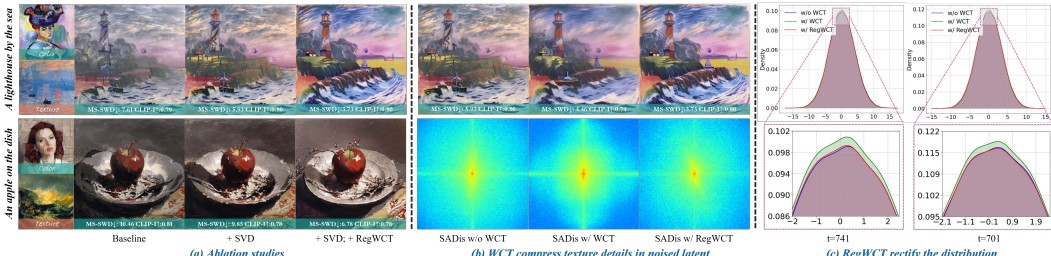

Figure 5: We extract texture information and suppress gray expression through SVD reweighting, and color alignment is improved by *RegWCT*.'Baseline' denotes color-texture disentanglement with Image-Prompt Additivity only (i.e., without SVD and *RegWCT*). (a) shows the Roles of SVD and *RegWCT* within *SADis*. (b) Our approach enables more precise color alignment with $\mathcal{I}_{clr}$ via *RegWCT*, whereas direct WCT manipulation compresses texture details and high-frequency components. (c) *RegWCT* rectifies the latent distribution that is affected by WCT. The distribution of the image latent generated with *RegWCT* technique is closer to that of the image generated without WCT in the frequency domain, across diverse time steps.

adding this difference to the embedding of another grey image, would transfer the color information to this image. This is illustrated for the figure of the grey bird image $C$. We can also see that the differences of $A - A'$ and $B - B'$ form a relatively compact distribution.

### 3.3.2 Color-Texture Extraction

Leveraging this image-prompt additivity, we disentangle color and texture representations from their respective reference images. For *color extraction*, we take the color image embedding and subtract the grayscale image embedding:

$$\mathbf{Emb}_{clr} = \tau_\phi(\mathcal{I}_{clr}) \ominus \tau_\phi(\mathbf{GS}(\mathcal{I}_{clr})) \tag{1}$$

where $\mathbf{GS}$ is the grayscale operation, $\mathbf{Emb}_{clr} \in \mathbb{R}^{n_t \times c}$. This strips away semantic information to retain only the color attributes (See Fig.4-b).

For *texture extraction*, we first convert the texture reference image to grayscale to remove any influence of its color palette, which is formulated as $\mathbf{Emb}_{tx}^* = \tau_\phi(\mathbf{GS}(\mathcal{I}_{tx}))$, $\mathbf{Emb}_{tx}^* \in \mathbb{R}^{n_t \times c}$. However, using only the grayscale texture embedding creates a mismatch in scale between the color and texture branches, often resulting in *overly gray tones* in T2I generations (as shown in Fig. 5 (a)-2nd col.). To address this, we average the gray texture image to obtain a pure gray image, which summarizes the main gray tone information in the $\mathbf{GS}(\mathcal{I}_{tx})$ image. Following that, we *concatenate* both embeddings from token-wise to obtain the initial texture representations:

$$\mathbf{Emb}_{tx}^{'} = \mathbf{Emb}_{tx}^* \copyright \tau_\phi(\mathbf{Avg}(\mathbf{GS}(\mathcal{I}_{tx}))) \tag{2}$$

$\mathbf{Emb}_{tx}^{'} \in \mathbb{R}^{2n_t \times c}$. Next, we conduct the Singular-Value Decomposition (SVD) over the texture representations. Inspired by [21, 44], we assume that the main singular values of $\mathbf{Emb}_{tx}^{'}$ correspond to the shared fundamental information of these two grayscale images, specifically the grayscale tone. We then have: $\mathbf{Emb}_{tx}^{'} = \boldsymbol{U\Sigma V}^T$, where $\boldsymbol{\Sigma} = diag(\sigma_0, \sigma_1, \cdots, \sigma_{n_t}, \cdots, \sigma_{2n_t-1})$,the singular values $\boldsymbol{\sigma}_0 \geq \cdots \geq \boldsymbol{\sigma}_{2n_t-1}$. To suppress the gray tone expression and extract the texture information, we introduce the augmentation for each singular value as:

$$\hat{\sigma} = \beta e^{-\gamma\sigma} * \sigma. \tag{3}$$

where $e$ is the exponential, $\gamma$ and $\beta$ are parameters with positive numbers. We then recover it as $\mathbf{Emb}_{re} = \boldsymbol{U\hat{\Sigma}V}^T$, with the updated $\hat{\boldsymbol{\Sigma}} = diag(\hat{\sigma_0}, \hat{\sigma_1}, \cdots, \hat{\sigma}_{2n_t-1})$. The texture embedding is obtained by $\mathbf{Emb}_{tx} = \mathbf{Emb}_{re}[: n_t, :]$. That effectively reduces residual gray-color influences while preserving texture details. After extracting both color and texture reference image embeddings, we achieve a baseline of the *DisIG* problem. The T2I inference process is now formulated as: $z_{t-1}^{'} = \mathcal{G}(z_t, t, \mathcal{C}_{text}, \mathbf{Emb}_{tx} \oplus \mathbf{Emb}_{clr})$. Note that, we only inject the disentangled embedding $\mathbf{Emb}_{tx} \oplus \mathbf{Emb}_{clr}$ into *the first decoder layer* to compute cross-attention maps, which shows better *stylization* performance as previous works proved [67, 68, 2, 66]. Examples shown in Fig. 6 (2nd cols.) demonstrate the effectiveness of our proposal.

### 3.3.3 Regularized Whitening-Coloring Transforms

**WCT: whitening-coloring transformation.** Solely applying our *CTE* method does not fully ensure a precise color palette match between the reference color image $\mathcal{I}_{clr}$ and the generated output (see Fig.5). We hypothesize that while CLIP embeddings capture high-level semantic information, they may not fully represent nuanced color distribution details. To better align the color distributions of the generated image with the color reference, we incorporate a whitening-coloring transform (WCT) [45, 9, 31] applied to the noisy latent as $z_t^{wct} = \mathbf{WCT}(z_t^{'})$ (detailed WCT formulas in the Supplementary). This approach allows for a more faithful color transfer by aligning the latent representations with the reference color palette, enhancing the stylistic consistency of generated images. Examples of applying the WCT transform during the T2I generation process are presented in Fig. 5-(b), demonstrating an improved alignment of color palettes between the generated images and the reference color images. This enhancement visually confirms the effectiveness of WCT in achieving more accurate color transfer, resulting in a closer match to the desired color distribution.

**Noise Regularization for WCT.** However, as shown in Fig. 5-(b), the WCT process tends to compress texture details, distorting high-frequency information, which is also observed in previous works [80, 42]. This issue is attributed to the signal-leak bias [16] and the inverted latent distribution gap from the inversion process [80]. These terms essentially describe the same underlying phenomenon: during training, the distribution of $z_T$—the latent representation after adding Gaussian noise to $z_0$—does not perfectly match a standard Gaussian. Instead, it always contains the leakage or some certain prior information towards the original image, causing the residual structure to persist even at high noise levels. This biased noise in the forward diffusion process leads to a slight misalignment in texture representation during generation. To address this, we propose adding a small-scale noise to the latent as $z_t = z_t^{wct} + \lambda \cdot \delta, \delta \sim \mathcal{N}(0,1)$, determined by a scale hyperparameter $\lambda$, to recover the lost high-frequency details. By introducing this noise regularization term, the regularized whitening-coloring transformation (*RegWCT*) achieves improved generative performance, better aligning both color and texture in the generated images with the reference images. Given a latent $z_t^{'}$ at timestep $t$, the color rectified latent $z_t$ after *RegWCT* is formulated as $z_t = (1 - \omega) z_t^{'} + \omega \cdot \mathbf{RegWCT}(z_t^{'})$, where $\omega$ acts as a balance weight. To prevent the inference trajectory from deviating too much from the original one, we only perform the *RegWCT* transformation during the intermediate steps as $[T_{start}, T_{end}]$. With all these techniques, including color-texture extraction (*CTE*) and Regularized WCT transformation (*RegWCT*), our method *SADis* achieves customizable and flexible Disentangled Stylized Image Generation (as shown in Fig. 5 (a)-4th col.).

### 3.3.4 ControlNet-based real image stylization

Our method can be integrated with ControlNet [79] $\mathcal{CN}$ to facilitate content-based stylized image generation, as illustrated in Fig. 1 (down) and Fig. 7 (a). By using any pretrained ControlNet model (e.g. Canny-conditioned) as the base pipeline while maintaining all other hyperparameters consistent with *SADis*, we are able to significantly broaden the applicability of our method. More specifically, we have an input $\mathbf{I}_c$ as the content image, it is passed through the ControlNet $\mathcal{CN}(\mathbf{I}_c)$ as conditions for T2I generation model as $z_{t-1}^{'} = \mathcal{G}(z_t, t, \mathcal{CN}(\mathbf{I}_c), \mathbf{Emb}_{tx} \oplus \mathbf{Emb}_{clr})$, where $z_T \sim \mathcal{N}(0,1)$ and the textual prompt as null $\mathcal{P} = $ "".

## 4 Experiments

### 4.1 Experimental Setups

**Datasets.** To ensure a fair comparison, we randomly select 40 images in total from the WikiArt [64] and StyleDrop [62] datasets. For our method, *SADis*, each of these images serves as either a color reference or a texture reference to enable color-texture disentanglement. In contrast, since the comparison methods lack disentanglement capabilities, we treat each of these 40 images as a style reference for these methods, supplemented by auxiliary text prompts generated from GPT-4o [1] as a strong captioning model. For quantitative comparisons, we used 20 images as the color reference set and 20 as the texture reference set. We also sampled 10 content prompts from StyleDrop, resulting in *4,000* stylized images per method to ensure fair and extensive comparisons with numerous images.

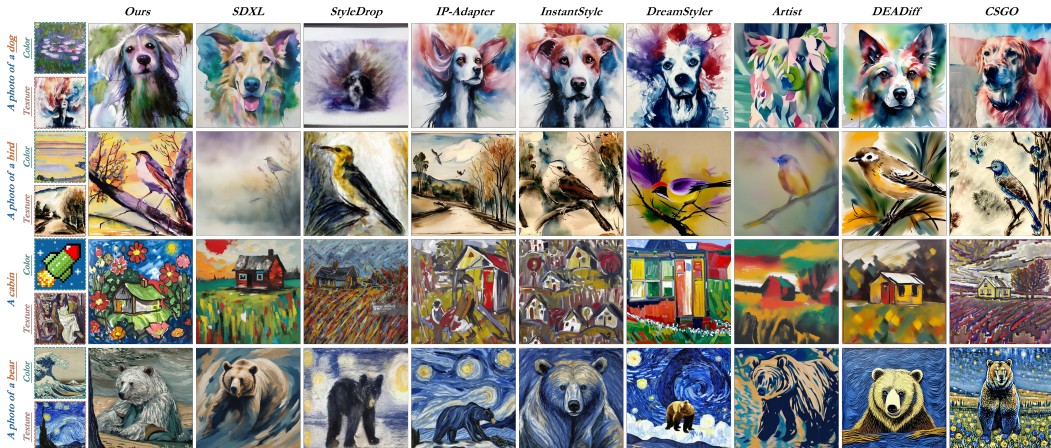

Figure 6: *Disentangled Stylized Image Generation* (*DisIG*) compared with baseline methods. For other approaches, we used GPT-4o to generate color descriptions based on the color reference image, incorporating these descriptions into the text prompts (details shown in the Supplementary). Our *SADis* accepts separate color and texture reference images to achieve flexible control.

**Evaluation Metrics.** We evaluate our method using multiple quantitative metrics. (1) For a general quality evaluation, we use CLIP-Score (CLIP) [28] to evaluate the T2I generation performance, specifically measuring the semantic consistency between the generated image and the text prompt. (2) For the evaluation of color attribute alignment, the MS-SWD [25] metric specifically evaluates color distance between generated and reference images. Color histogram distance (C-Hist) computes the distance between both color histograms. The GPT-4o color score [1] requests the GPT model to rate from 0-5 according to the color consistency between the color reference and the generated image. (3) For the texture quality, the Kernel-Inception Distance (KID) [5] is assessing the quality of generated images by measuring the dissimilarity between the real and generated image distributions (We use feature dimension as 2048 and evaluate on the 4000 images). CLIP-I is used to evaluate the similarity between the texture image and the generated image in grayscale.

**Implementation Details.** We build our method *SADis* upon the SDXL [53]. To apply the image embeddings as additional conditions to the T2I model, we use the IP-Adapter [76] pretrained projectors. Both of these models are based on the CLIP model [55], where the SDXL model utilizes the CLIP text encoder to generate textual embeddings and the IP-Adapter leverages the CLIP image encoder to extract image embeddings before the embedding projector. For the hyperparameters, we set $\lambda = 0.01, T_{start} = 0.8T, T_{end} = 0.6T, \omega = 0.5, \gamma = 0.003, \beta = 1.0$. For the time cost, we reported for 50-step DDIM. All the experiments are conducted on a single L40s GPU.

**Comparison Methods.** We compare with several state-of-the-art tuning-free stylization approaches based on the T2I diffusion models: DEADiff [54], InstantStyle [67], IP-Adapter [76], CSGO [74], DreamStyler [3], StyleDrop [62] and Artist [39]. For these baseline methods, we utilize the GPT-4o [1] model to generate captions from the color reference image, which are appended to the original textual prompt to guide the T2I generation. Additionally, SDXL [53] and Artist model are included in comparisons by generating captions from both the color and texture reference images, appending these captions to the textual prompts to guide the generation process. *We include much more experimental setup details, qualitative and quantitative results in the Supplmentary Material.*

## 4.2 Experimental Results

**Qualitative Comparison.** In our comparison of generation quality, visual results under the Disentangled Stylized Image Generation (*DisIG*) scenario are shown in Fig. 6 and Fig. 7. While baseline methods manage to capture some style information from the style reference image (the texture image for *SADis*), they fail to represent the color palette provided by the color reference accurately. The IP-Adapter and InstantStyle methods even retain original layouts and elements from the style images, deviating from the intended stylization. Other baselines, such as Artist [39] and DEADiff [54], exhibit poor generation quality, reflecting a limited understanding of both color and texture attributes. In contrast, our method, *SADis*, allows for both texture and color reference inputs, offering precise, con-

Table 1: Quantitative Comparison with existing image stylization methods. The best and second-best numbers are marked with **bold** and underlined respectively.

| Method | CLIP | Color | | | Texture | | Time Cost (s) | User study (%) | | |
|---|---|---|---|---|---|---|---|---|---|---|
| | | MS-SWD↓ | C-Hist↓ | GPT4o↑ | CLIP-I↑ | KID↓ | | Color↑ | Texture↑ | Both↑ |
| SDXL [53] | 0.272 | 9.51 | 1.20 | 3.01 | 0.69 | 0.08 | 9.26 | 16.99 | 5.84 | 11.65 |
| IPAdapter [76] | 0.233 | 11.54 | 1.23 | 2.84 | **0.84** | **0.043** | 9.52 | 6.08 | 22.54 | 14.55 |
| InstantStyle [67] | 0.261 | 12.53 | 1.32 | 2.80 | 0.74 | 0.056 | 9.31 | 4.81 | 24.94 | 13.10 |
| Artist [39] | 0.269 | 10.48 | 1.39 | 2.82 | 0.69 | 0.089 | 12.32 | 9.38 | 1.79 | 3.40 |
| DEADiff [54] | 0.267 | 11.20 | 1.24 | 2.73 | 0.69 | 0.087 | **1.86** | 4.57 | 2.59 | 2.67 |
| StyleDrop [62] | 0.275 | 13.52 | 1.43 | 2.67 | 0.70 | 0.054 | 6.91 | 5.07 | 3.57 | 5.34 |
| DreamStyler [3] | 0.277 | 12.17 | 1.26 | 2.39 | 0.71 | 0.060 | 5.23 | 4.67 | 3.57 | 5.39 |
| CSGO [74] | 0.280 | 14.25 | 1.36 | 2.63 | 0.69 | 0.071 | 15.99 | 6.59 | 9.73 | 6.06 |
| *SADis* (Ours) | **0.281** | **5.57** | **0.96** | **3.34** | 0.74 | 0.049 | 10.30 | **41.83** | **25.42** | **37.84** |

Table 2: Ablation by removing each components in our method *SADis*.

| Method | Color | | Texture | Time cost (s) |
|---|---|---|---|---|
| | MS-SWD↓ | C-Hist↓ | CLIP-I↑ | |
| *SADis* (Ours) | **5.57** | **0.96** | 0.74 | $\approx 10.30$ |
| − SVD | 5.70 | 1.01 | **0.76** | $\approx 10.29$ |
| − *RegWCT* | 8.05 | 1.06 | 0.75 | $\approx 9.42$ |
| − SVD − *RegWCT* | 8.93 | 1.10 | **0.76** | $\approx 9.41$ |

trollable T2I stylization. The generated images by *SADis* exhibit significantly improved color fidelity and texture detail compared to other approaches, demonstrating its effectiveness in disentangled style attribute transfer. To further validate *SADis*'s robustness, we conduct two tests: (1) swapping the color and texture reference images as shown in Fig. 7 (c), and (2) modifying their saturation and illumination levels, as shown in Fig. 7 (d). Unlike existing approaches such as InstantStyle, which cannot preserve image content under these conditions (detailed in *Supplementary*), *SADis* successfully applies color and texture independently while keeping the content largely unchanged.

**Quantitative Comparison.** A detailed quantitative comparison is provided in Tab. 1 to further support our findings. Our method, *SADis*, retains text-image alignment quality at a level comparable to the base SDXL model [53], as indicated by the CLIP score. This comparison demonstrates that *SADis* preserves alignment with the textual prompt better than other methods. Regarding color alignment, *SADis* significantly outperforms other approaches. For texture representation, the IP-Adapter-based methods, including InstantStyle [67] and *SADis*, show superior performance over other baselines. They achieve higher texture-related metrics at the expense of color fidelity and text-image alignment quality. The high texture scores for IP-Adapter can be explained by the fact that they control all cross-attention layers; However, this leads to high-semantic content leakage (like the same stars in the Van Gogh bear in Fig. 6). We decided to prevent this by only controlling the cross-attention of the low-level layers following InstantStyle [67, 2, 66]; Therefore, our texture results are close to those of InstantStyle, but at the same time we greatly improve color fidelity. In addition, their generated images follow the same structure as the original texture image, showing that they do not disentangle structure from texture but overfit to the contents of the texture images.

**Ablation Study.** The ablation study for each component of *SADis* is listed in Tab. 2. Results show that SVD rescaling in the *CTE* process and the noise-regularized *RegWCT* techniques significantly enhance color alignment while only slightly reducing texture precision, which is almost undetectable in the CLIP-I metric. The trade-off between texture and color alignment is optional for users, allowing them to adjust the balance based on the requirements.

**User Study.** To better assess alignment with human preferences, as shown in Tab. 1(right), we conducted a user study with 24 participants (30 sextuplets/user), collecting 720 data for each method. Each was asked to "select the best image from pairs of generated images, taking into account overall quality (considering *both* color and texture alignments), color alignment, and texture alignment, respectively." Our method outperformed other baseline approaches with at least a 20% improvement. This demonstrates the potency of *SADis* and its high alignment with human preference.

## 4.3 Additional Features

**Image-based stylization.** *SADis* Integrated with ControlNet achieves image-based stylized generation, as shown in Fig. 7 (a). Here, we adopt ControlNet (Canny) as the base model, setting its conditioning scale to 0.6. This integration broadens the application scenarios of *SADis*.

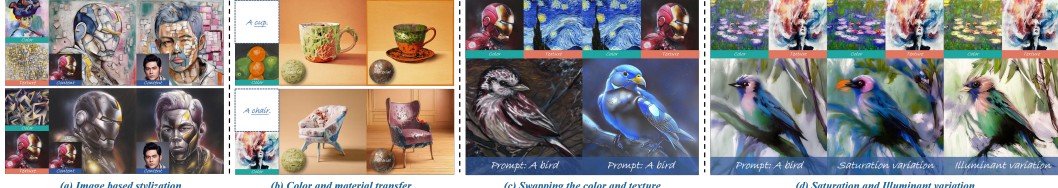

*(a) Image based stylization*  *(b) Color and material transfer*  *(c) Swapping the color and texture*  *(d) Saturation and Illuminant variation*

Figure 7: (a) *SADis* allows for *real-image stylization*. (b) *SADis* effectively disentangles **color** and **material** elements from separate images, enabling precise control over color and material in image generation. (c) *SADis* remains effective when exchanging texture and color images. (d) *SADis* can capture color image properties such as saturation and illuminant. This further demonstrates the robustness of *SADis* in realistic *DisIG* scenario.

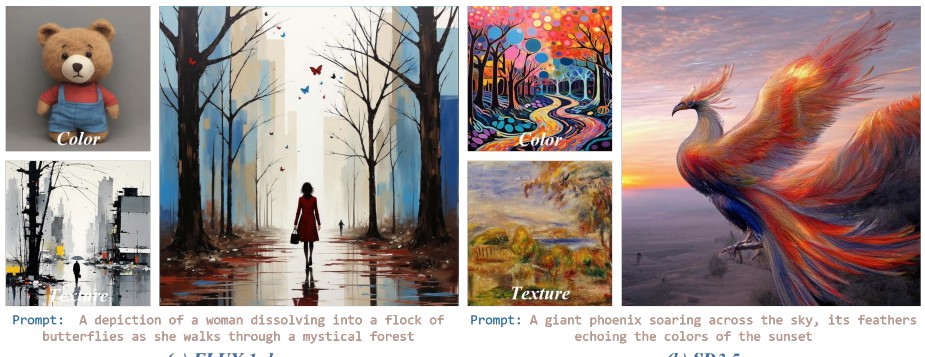

Prompt: A depiction of a woman dissolving into a flock of butterflies as she walks through a mystical forest
Prompt: A giant phoenix soaring across the sky, its feathers echoing the colors of the sunset

*(a) FLUX.1-dev*  *(b) SD3.5*

Figure 8: SADis is compatible with MMDiT-based models. (a) demonstrates color-texture disentangled stylization results on FLUX.1-dev, while (b) shows the corresponding results on SD3.5.

**Color and material transfer.** Given color and material references from separate images, *SADis* is also capable of disentangling color and material elements, and provides precise controls over color and materials in T2I generation. The color and material transfer results are shown in Fig. 7 (b). This capability of *SADis* highlights its potential applications in artistic creation and industrial design.

**Compatible with MMDiT-based models.** *SADis* is also compatible with MMDiT-based models (SD3.5 [14] and FLUX.1-dev[7]) to achieve color-texture disentangled stylization. Specifically, we use SD3.5 and FLUX.1-dev as the base models. These models employ the original text encoders (i.e., T5, CLIP-L, and CLIP-G) for content control. We utilize SigLIP [78] with a pretrained adapter[8,9] for style control. Color-texture embeddings are extracted and disentangled from SigLIP features, following the same workflow as illustrated in Fig. 3. As shown in Fig. 8, SADis achieves color-texture disentanglement and enables stylized image generation on these MMDiT-based models as well. This demonstrates that the color-texture disentanglement capability of SADis does not rely on the U-Net architecture and can be generalized to MMDiT-based models for stylized image generation. It also indicates that SADis is not limited to specific CLIP models and can be applied to other text-image pretrained encoders.

## 5 Conclusion

We addressed a critical challenge in stylized image generation by introducing the concept of disentangled stylized image generation (*DisIG*). We focused on color and texture as key style attributes, presenting the first approach, named style attribute disentanglement (*SADis*), for independent control over these elements. Using the image-prompt additivity property, we proposed novel techniques, including the color-texture extraction (*CTE*) and regularized whitening-coloring transformation (*RegWCT*), to ensure enhanced color-texture consistency and more accurate results. Experimental evaluations demonstrate that *SADis* significantly outperforms existing stylization methods, both qualitatively and quantitatively. This work also opens new avenues for more flexible and customizable image generation, paving the way for future innovations for art creators.

---

[7]https://huggingface.co/black-forest-labs/FLUX.1-dev.

[8]https://huggingface.co/InstantX/SD3.5-Large-IP-Adapter.

[9]https://huggingface.co/InstantX/FLUX.1-dev-IP-Adapter.

## Acknowledgements

We acknowledge project PID2022-143257NB-I00, financed by MCIN/AEI/10.13039/501100011033 and ERDF/EU, and the Generalitat de Catalunya CERCA Program, and ELLIOT project funded by the European Union under Grant Agreement 101214398. This work was also supported by NSFC (NO. 62225604) and Youth Foundation (62202243). We acknowledge "Science and Technology Yongjiang 2035" key technology breakthrough plan project and Chinese government-guided local science and technology development fund projects (scientific and technological achievement transfer and transformation projects) (254Z0102G). Kai Wang acknowledges the funding from Guangdong and Hong Kong Universities 1+1+1 Joint Research Collaboration Scheme and the start-up grant B01040000108 from CityU-DG.

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

# Appendix

## A Statements

**Limitations.** The present study focuses on disentangling the color and texture elements in a training-free manner, and performing stylized image generation using these elements. This method can offer a flexible and customizable image generation for art creators and designers. However, when applying to content-consistency stylization, the consistency generation ability of this work can be further improved. In our future work, we will explore deeper with the consistency and expand our research into content-consistency stylization.

**Broader Impacts.** *SADis* enhances the flexible stylization capability in text-to-image synthesis by disentangling the color and texture elements. However, it also carries potential negative implications. It could be used to generate false or misleading images, thereby spreading misinformation. If *SADis* is applied to generate images of public figures, it poses a risk of infringing on personal privacy. Additionally, the automatically generated images may also touch upon copyright and intellectual property issues.

**Ethical Statement.** We acknowledge the potential ethical implications of deploying generative models, including issues related to privacy, data misuse, and the propagation of biases. All models used in this paper are publicly available. We will release the modified codes to reproduce the results of this paper. We also want to point out the potential role of customization approaches in the generation of fake news, and we encourage and support responsible usage.

**Reproducibility Statement.** To facilitate reproducibility, we will make the entire source code and scripts needed to replicate all results presented in this paper available after the peer review period. We will release the code for the novel color metric we have introduced. We conducted all experiments using publicly accessible datasets. Elaborate details of all experiments have been provided in the Appendices.

## B Image-Prompt Additivity

### B.1 Broader Property of Image-Prompt Additivity

We demonstrate the broader property of Image-Prompt Additivity in Fig. 9. For instance, in Fig. 9-(a), subtracting the embedding of a hat from the embedding of a person wearing a hat results in the generation of a person without a hat. Similarly, in Fig. 9-(b), adding the embedding of glasses to a person results in the generation of that person wearing glasses. We refer to this phenomenon as Image-Prompt Additivity. We hypothesize this property originates from the image-text paired training in CLIP models [55]. The training process endows the image branch with the additivity property inherent to the text embedding space [50, 33, 6]. Although the person identities are altered after embedding additivity manipulations, this approach demonstrates a promising property for enabling our training-free color-texture disentanglement. Building on this property, we develop our method, *SADis*, to extract color and texture information from reference images and effectively apply these attributes in T2I generation.

In future work, we aim to address the limitations of Image-Prompt Additivity by enhancing identity consistency after additivity manipulations. This could broaden the impact and applicability of this property, advancing its utility in the field of image generation.

### B.2 Extra Analysis on Image-Prompt Additivity

As another illustration of image prompt additivity, we construct a set of blue, red, and green images. For example, the blue set is constructed by setting the red and green color channels to zero of a set of 100 images (similarly, for the green and red set). We isolate the *color representation* by performing a feature subtraction between the blue image and its grayscale equivalent. We plot their projected embeddings as blue dots in Fig. 10-(Right); note how these embeddings always maintain close to the pure blue embedding and are far from the green and red embeddings. Also, generated images with these subtraction blue color embeddings keep the bluish color palette as expected (see the car and t-shirt). An extended analysis is presented in Fig. 11, where we apply similar manipulations using

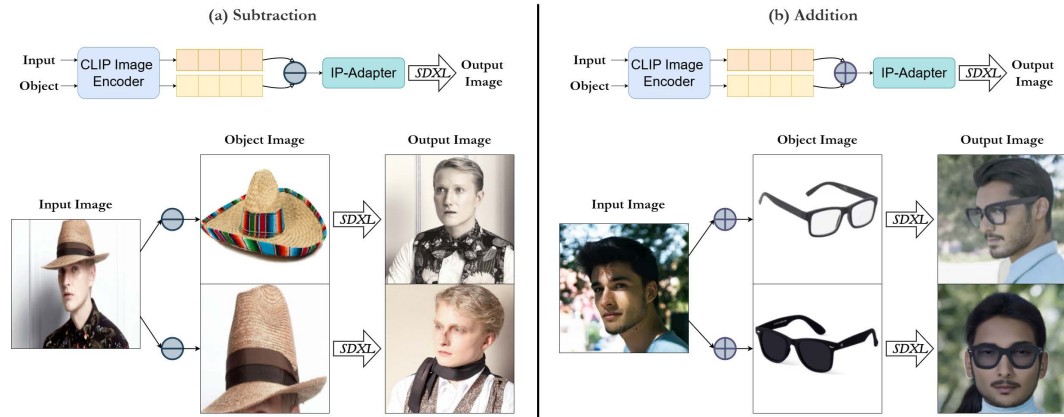

Figure 9: By subtracting or adding the object image embeddings from the input image embeddings, we can effectively remove or add the object to the scene, although some degree of identity information for the person is also diminished.

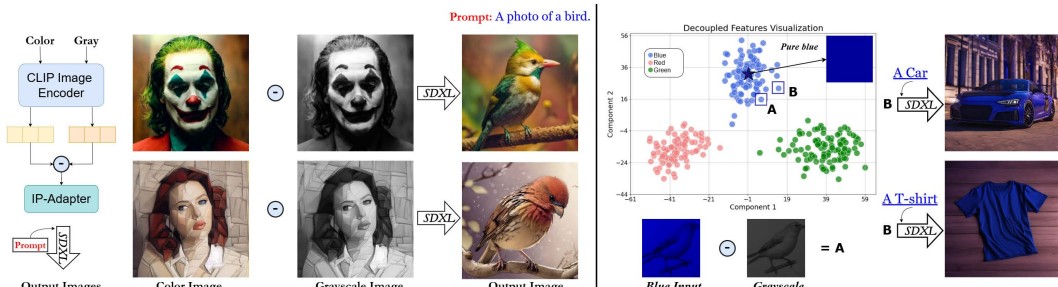

Figure 10: (Left) By subtracting the grayscale image embeddings from the color image embeddings, we can effectively remove the texture information and isolate the color information. Combined with the prompt "A photo of a bird" we can generate images in the same color schemes as the input images. (Right) By subtracting the bluish image embeddings with the grayscale embeddings for random generated images with 100 prompts, we visualize the subtracted embedding via PCA decomposition. We observe them gathering around the pure blue image. The generation with these subtracted embeddings further prove the consistency.

colorful images as color references. The successful clustering further confirms that the resulting embeddings effectively preserve essential color information.

## C  Implementation details

We develop our method, *SADis*, based on the SDXL model [53], which is among the leading open-source T2I generative models available. To construct the experimental datasets, we randomly select 40 images from the WikiArt dataset [64] and StyleDrop [62] image collections. Each image can be used either as a color reference or a texture reference in the experiments. All input images are resized to $512 \times 512$ before feeding into models. Also, for quantitative comparisons, 20 images are designated as the color reference set, while the remaining 20 serve as the texture reference set. Additionally, we sample 10 content prompts from StyleDrop, yielding a total of 4000 stylized images for each method for comparison. Since some comparison methods (such as SDXL [53] and Artist [39]) require text prompts as style controls for image generation, we employ the state-of-the-art vision-language model GPT-4o as the image captioning tool to generate precise color and texture prompts from the reference images. Subsequently, the content, color, and texture prompts are concatenated in the format: *'{content prompt}, {texture prompt} in the color {color prompt}'*. As for the comparison models (like DEADiff [54], IP-Adapter [76], Instantstyle [67], DreamStyler [3], StyleDrop [62], and CSGO [74]) that require text prompts for color control, the text prompt is constructed as: *'{content prompt}, in the color {color prompt}'*. All experiments were conducted on an NVIDIA-L40s GPU.

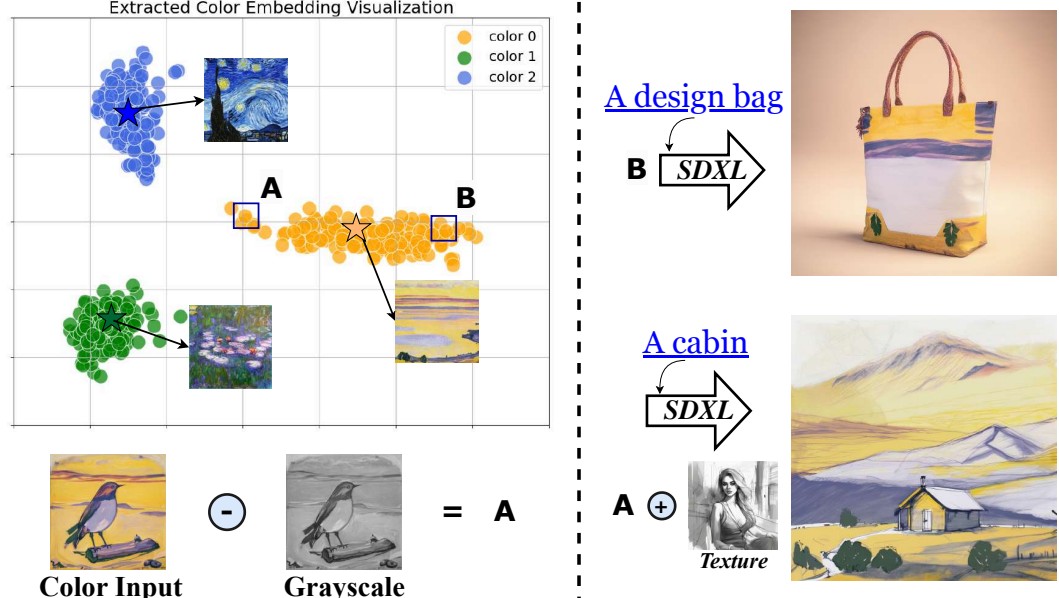

Figure 11: By subtracting the color image embeddings with the grayscale embeddings for random generated images with 100 prompts, we visualize the subtracted embedding via PCA decomposition. We observe them gathering around the color reference image. The generation with these subtracted embeddings further prove the consistency.

We further include the details of each evaluation metric. **CLIP Score** [55] is used to evaluate the semantic alignment between the text prompt and the generated image. We calucate the CLIP Score according to configuration of the T2I-CompBench repository [34]. For the color alignment evaluation, we use MS-SWD [25], color histogram distance (C-Hist), and GPT4o to calculate the color attribute similarity between the color reference and the generated image. **MS-SWD** [25]: since the generated imgae are usually not spatially aligned with the color reference image, we use MS-SWD to better evaluate the color attribute alignment according to the default setting of their repository [25]. **C-Hist:** we first compute the RGB histograms of the color reference and generated images. Afterwards, the Bhattacharyya distance is used to measure the differences between their color histograms. **GPT-4o score** [1]: to comprehensively evaluate the color alignment performance, we furture adopt the multimodal model GPT-4o to compute the color alignment score between the generated image and the color reference image. Specifically, the GPT-4o metric is computed according to Fig. 12. Firstly, following the previous work[17], we extract dominant colors by ColorThief. Afterwards, we feed the extracted color names to GPT-4o as the reference color from the color image, and ask GPT-4o to give 1-5 points according to the criteria shown in Fig. 12.

## D  WCT transformations formulas

Given a latent $z_t^{'} \in \mathcal{R}^{C \times H \times W}$ and a color reference latent $z_t^c \in \mathcal{R}^{C \times H \times W}$ at timestep $t$, we adopt Whitening and Coloring Transforms (WCT) [45, 9, 31] to transform $z_t^{'}$ to match covariance matrix of color reference latent $z_t^c$. There are two step for WCT: **Whitening transform** and **Coloring transform**.

**Whitening transform:** the latent $z_t^{'}$ is firstly centered by subtracting its mean vector $m$, and then an uncorrelated latent $\hat{z}_t^{'}$ is obtained by:

$$\hat{z}_t^{'} = ED^{-1/2}E^{\mathrm{T}}z_t^{'}, \tag{4}$$

where $D$ denotes a diagonal matrix of $z_t^{'}z_t^{'\mathrm{T}} \in \mathcal{R}^{C \times C}$ and $E$ is the corresponding orthogonal matrix of eigenvectors which satisfy $z_t^{'}z_t^{'\mathrm{T}} = EDE^{\mathrm{T}}$.

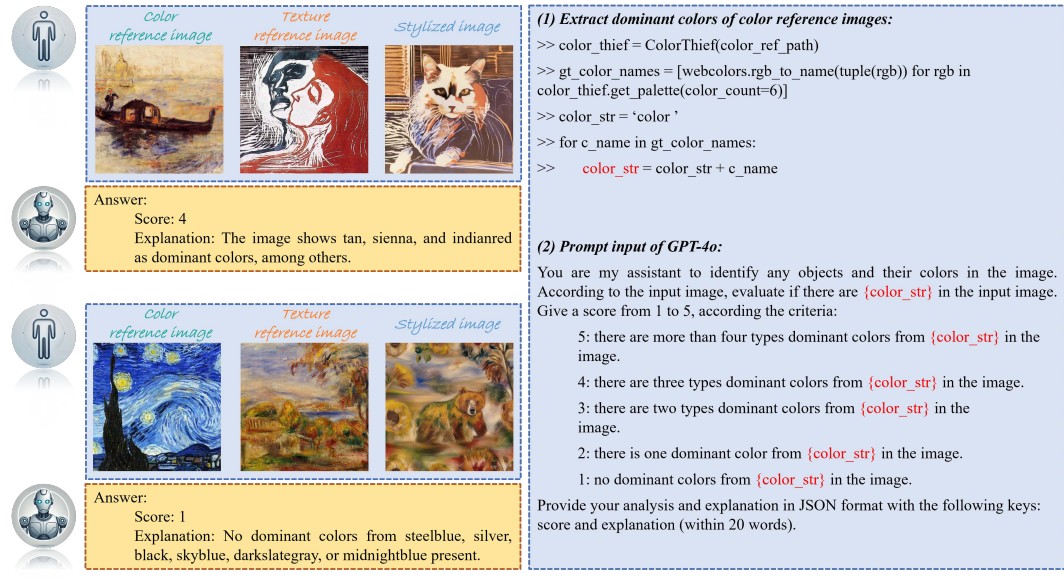

Figure 12: GPT-4o metric for color alignment evaluation.

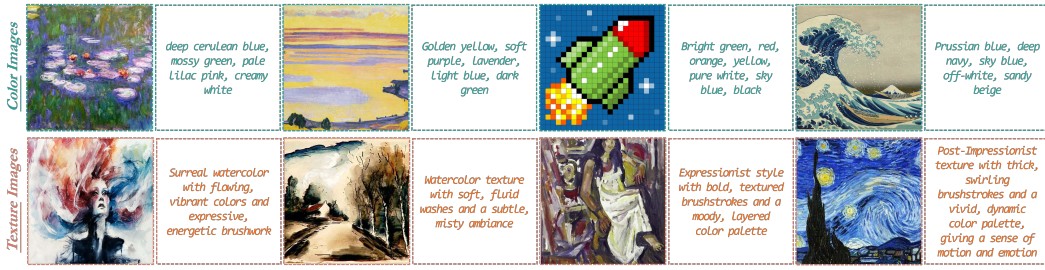

Figure 13: Some color and texture prompts generated by GPT-4o for comparison methods in the main paper for qualitative evaluation.

**Coloring transform:** it's the reverse process of Whitening transform [32, 45]. Beforehand, the color reference latent $z_t^c$ is centered by subtracting its mean vector $m_c$. Subsequently, we obtain the transformed latent $z_t^{wct}$ which satisfies the desired correlations $z_t^{wct} z_t^{wct\mathrm{T}} = z_t^c z_t^{c\mathrm{T}} = I$:

$$z_t^{wct} = E_c D_c^{-1/2} E_c^{\mathrm{T}} \hat{z}_t^{'}, \tag{5}$$

where $D_c$ is a diagonal matrix with the eigenvalues of the covariance matrix $z_t^c z_t^{c\mathrm{T}}$ and $E_c$ is the corresponding orthogonal matrix of eigenvectors. Finally, we re-center the WCT transformed latent $z_t^{wct}$ by adding the mean vector $m_c$ of the color reference latent $z_t^c$:

$$z_t^{wct} = z_t^{wct} + m_c. \tag{6}$$

# E  Additional Experimental Results

In Fig. 14, we include more color-texture disentanglement examples of our method *SADis*, which is under the Disentangled Stylized Image Generation (*DisIG*) scenario.

**Additional comparison results with complex prompts.** To further evaluate performance with complex prompts, we conducted additional experiments by randomly sampling 10 complex prompts from DREAMBENCH++ [52], together with 10 color and 10 texture images from our dataset, resulting in 1000 generated images for evaluation. As shown in Tab. 3, compared to other methods,

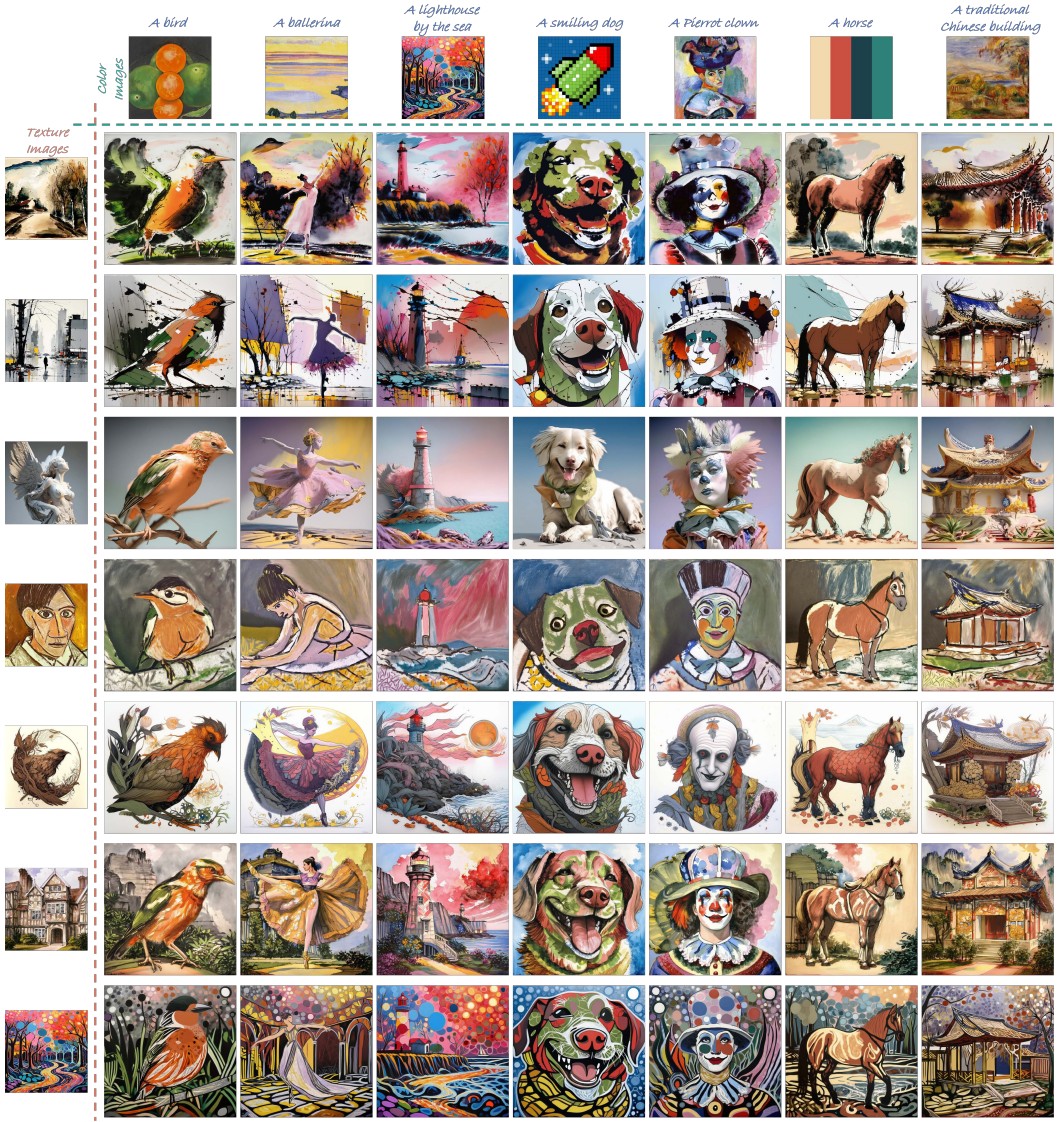

Figure 14: Additional experimental results of *SADis*.

our approach achieves the best disentanglement of color and texture, resulting in a more balanced performance in both color and texture consistency. Specifically, our method attains significantly higher color consistency, while other methods often exhibit strong color-texture entanglement, with color being overly influenced by the texture reference image. Although IP-Adapter achieves the highest numerical score for texture consistency, it suffers from severe semantic leakage from the texture reference image (Row 2 and Row 3 in Fig. 6), leading to poor text-image alignment (CLIP score: 0.257), which is inadequate for stylization. In contrast, except for IP-Adapter, our method achieves the highest scores in both texture consistency and text-image alignment.

**Image-based stylization.** Furthermore, our method, *SADis*, can be seamlessly integrated with ControlNet [79] to enable image-based stylized generation and material transfer, as demonstrated in Fig. 16 and Fig. 15(the last row) respectively. Here, we adopt ControlNet (Canny) as the base pipeline, setting its conditioning scale to 0.6. All other hyper-parameters are kept consistent with those of *SADis*. This integration broadens the application scenarios of our proposed approach.

**Color and material transfer.** Given color and material references from separate images, our method, *SADis*, is also capable of disentangling color and material elements, and provides precise controls over color and materials in T2I generation. The color and material transfer results are shown in

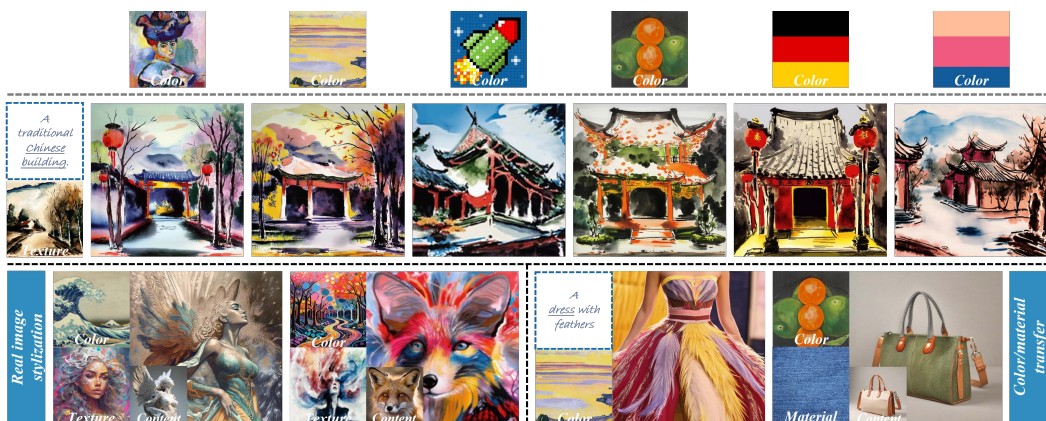

Figure 15: Stylized images generated by our *training-free* method, *SADis*. (Up) As shown in the first rows, it enables disentangled control over *color* and *style* attributes in text-to-image diffusion models using separate image prompts. This approach offers creators enhanced color control, including the use of color palettes as in the last two columns. (Down) *SADis* also enables real-image stylization by incorporating a content image as an additional condition via ControlNet. Furthermore, it extends to color-only stylized generation and material transfer for more flexible image generation.

Table 3: Quantitative comparison with complex prompts. The complex prompts are sampled from DREAMBENCH++ [52]. The best and second-best numbers are marked with **bold** and underlined respectively.

| Method | CLIP | Color | | | Texture | |
|---|---|---|---|---|---|---|
| | | SWD↓ | C-Hist↓ | GPT4o↑ | CLIP-I↑ | KID↓ |
| SDXL [53] | 0.291 | 9.00 | 1.14 | 3.07 | 0.698 | 0.091 |
| IPAdapter [76] | 0.257 | 9.99 | 1.15 | 2.96 | **0.817** | **0.058** |
| InstantStyle [67] | 0.278 | 11.21 | 1.25 | 2.85 | 0.751 | 0.065 |
| Artist [39] | 0.253 | 11.15 | 1.24 | 2.67 | 0.742 | 0.082 |
| DEADiff [54] | 0.280 | 10.01 | 1.14 | 2.80 | 0.718 | 0.090 |
| StyleDrop [62] | 0.290 | 11.90 | 1.35 | 2.83 | 0.731 | 0.078 |
| DreamStyler [3] | 0.300 | 11.27 | 1.21 | 2.57 | 0.677 | 0.083 |
| CSGO [74] | 0.299 | 12.60 | 1.22 | 2.64 | 0.728 | 0.081 |
| *SADis* (Ours) | **0.301** | **6.08** | **0.95** | **3.14** | 0.751 | 0.064 |

Fig. 19 and Fig. 15(the last row). This capability of *SADis* highlights its potential applications in artistic creation and industrial design. We also desire to note that, the term "color palette" typically refers to the overall color scheme of an image, not specific local regions. Our *SADis* does not control the color palette of individual objects in general. However, Fig. 19 demonstrates controlled color and material transfer using a masking mechanism with our method *SADis*.

**Color and texture transfer from the same image.** *SADis* performs exceptionally well when using the same image as both the color and texture reference (as shown in Fig. 20), showcasing its remarkable flexibility and adaptability. This ability highlights the method's robustness in leveraging a single source to effectively guide both color and texture information, ensuring consistent and coherent results. The comparative experimental results are presented in Tab. 4. In terms of texture consistency, our method achieves the best performance among all methods except for IP-Adapter. However, it is important to note that although the IP-Adapter achieves the highest numerical score for texture consistency, it introduces severe semantic leakage from the texture reference image (see Row. 2 and Row. 3 of IP-adapter results in Fig. 6), resulting in poor text-image alignment (CLIP: 0.260 in Tab. 4), which fails to meet the requirements of stylization. In contrast, excluding the IP-Adapter, our method achieves the highest scores in color consistency (MS-SWD: 3.19), texture consistency (CLIP-I: 0.754), and text-image alignment (CLIP: 0.302).

**Robust to color variations.** As shown in Fig.21 and Fig. 22, we vary the saturation and illuminant continuously. To be specific, for saturation, we developed a Python-based saturation adjustment tool

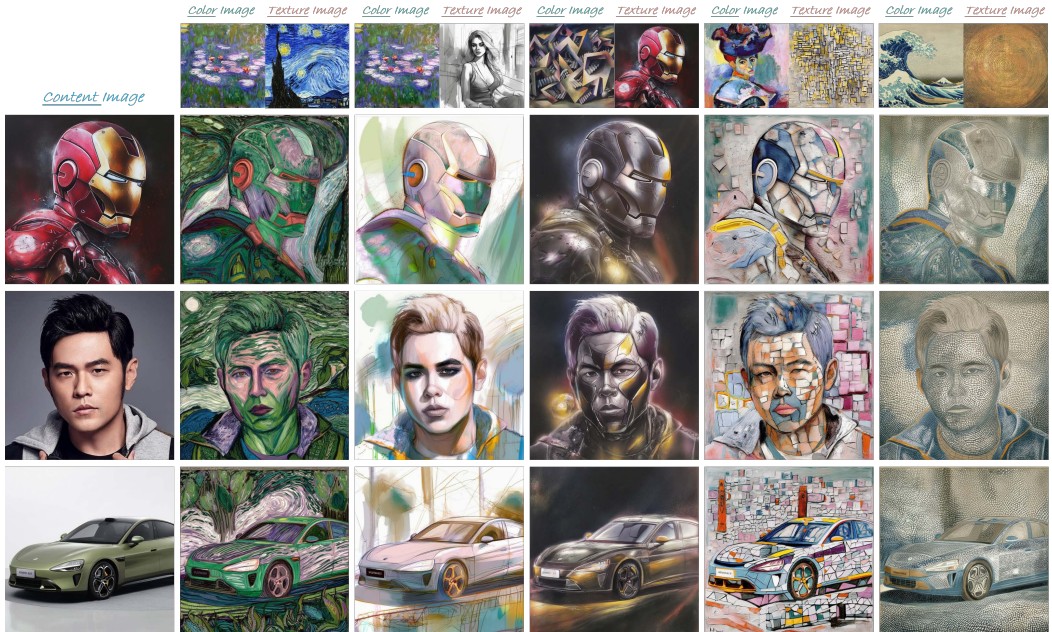

Figure 16: Our work is compatible with ControlNet to achieve **image-based stylization**.

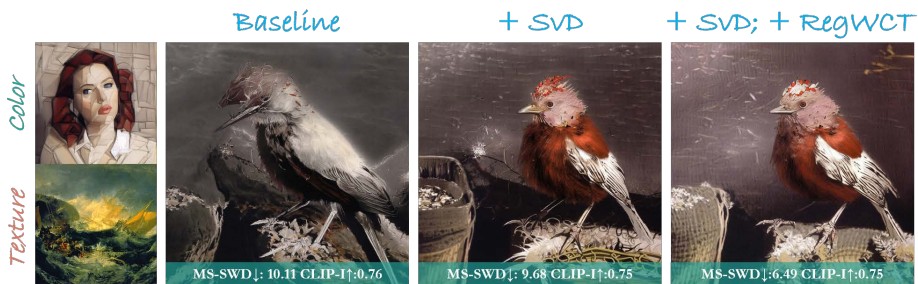

Figure 17: Visualization of ablating each component of *SADis*.

utilizing HSV (Hue, Saturation, Value) color space transformation. The tool linearly modifies the saturation channel using factors of [0.2, 0.6, 1.0, 1.5, 3.0], respectively, before converting the images back to RGB format. The results of saturation variations are shown in Fig. 21. Regarding the color of the illuminant, we modify images by applying calibrated RGB channel multipliers. For warm temperatures (red) as shown in Fig. 22, red increases by 30%, green reduces by 10%, and blue by 20%. The intensity of these adjustments can be controlled (0-1 range), enabling fine-grained color temperature manipulation. We present 3 increasing intensities in Fig. 22. Our method *SADis* is able to generate images with smooth change along with the saturation and illuminant color variations, while the other method InstantStyle [67] fails. That further proves the robustness of our method *SADis* for Disentangled Stylized Image Generation (*DisIG*).

**Each component works independently.** The T2I generations in Fig. 18-left provide evidence that the color and texture branches in *SADis* function independently. Additionally, the visualization for our ablation study in Fig. 17 further demonstrates improved color alignment with minimal texture degradation, achieving a good trade-off. However, the trade-off between texture and color alignment is optional for users, allowing them to adjust the balance based on the requirements of specific application scenarios.

**Stability of different sampling rounds.** We conducted additional experiments to assess the stability of color and texture in repeated generations. For each setting, the process was repeated five times, generating 1,000 images per round and computing the relevant metrics. As summarized in Tab. 5, the results show minimal variation in color and texture consistency across rounds, demonstrating the robustness and stability of our method.

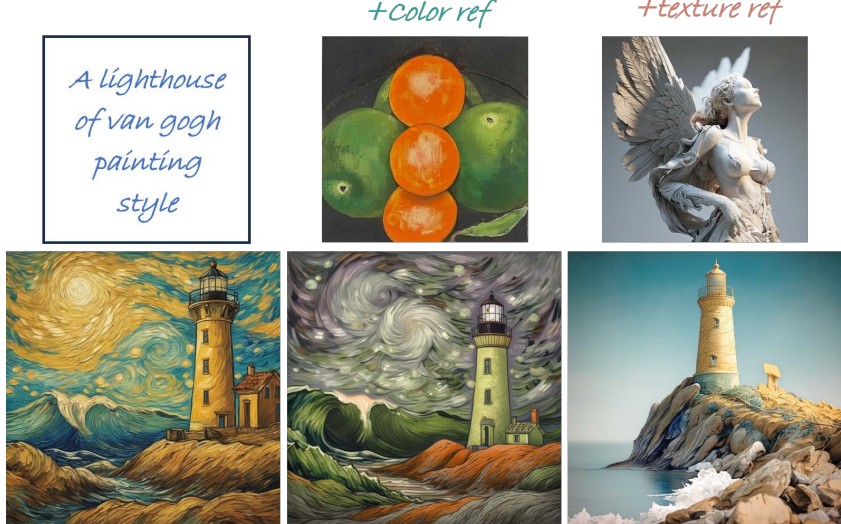

Figure 18: *SADis* can control color and texture generation separately using reference images.

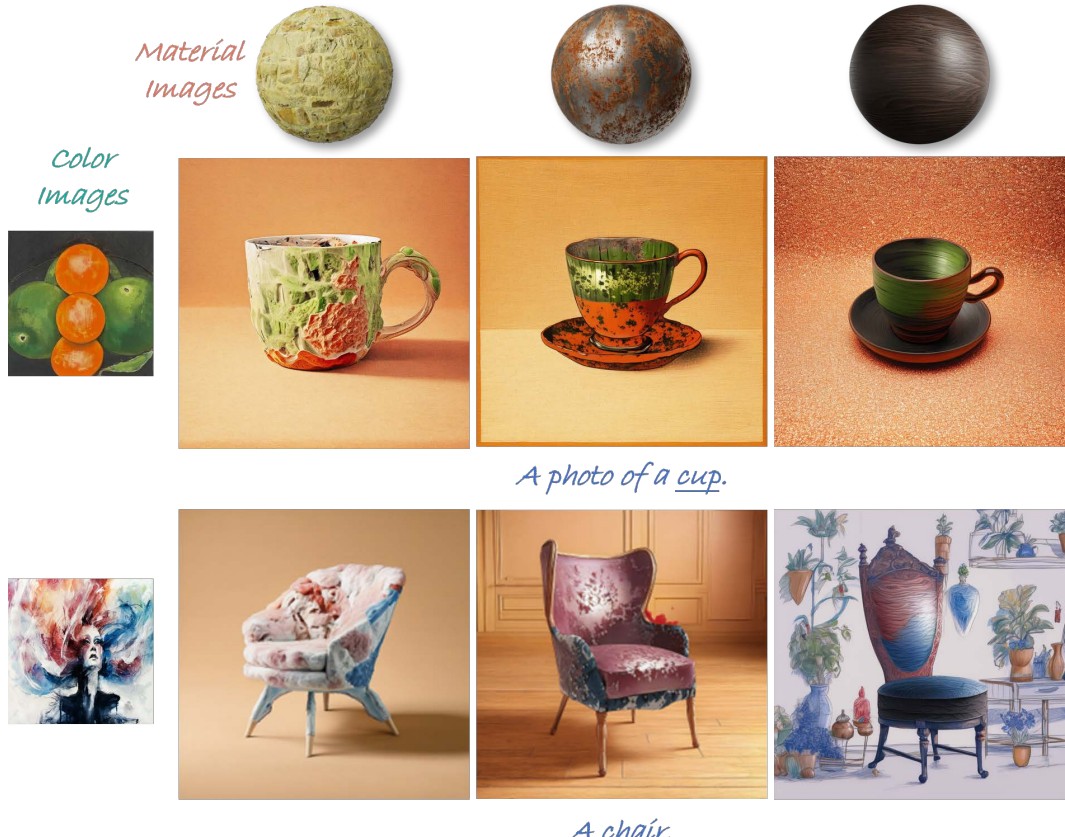

Figure 19: *SADis* effectively disentangles **color** and **material** elements from separate images, enabling precise control over color and material in image generation.

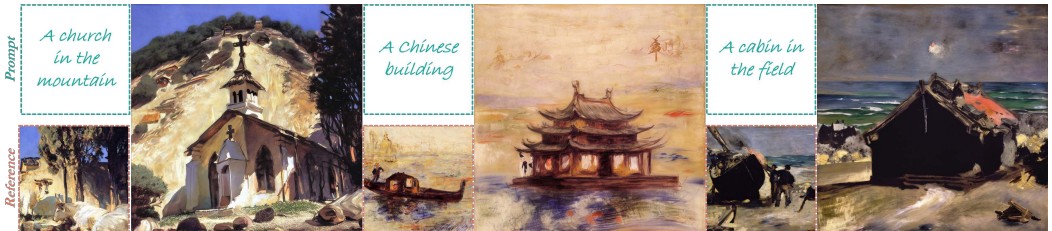

Figure 20: *SADis* performs well using the same image as both the color and texture reference, demonstrating its flexible capability.

Table 4: Quantitative comparison where both color and texture are derived from the same image. The best and second-best numbers are marked with **bold** and underlined, respectively.

| Method | CLIP | Color | | | Texture | |
| --- | --- | --- | --- | --- | --- | --- |
| | | MS-SWD↓ | C-Hist↓ | GPT4o↑ | CLIP-I↑ | KID↓ |
| SDXL [53] | 0.292 | 9.12 | 1.15 | 3.04 | 0.696 | 0.090 |
| IPAdapter [76] | 0.260 | 4.18 | 0.63 | 3.44 | **0.815** | **0.057** |
| InstantStyle [67] | 0.277 | 4.21 | 0.64 | 3.41 | 0.753 | 0.066 |
| Artist [39] | 0.272 | 10.98 | 1.18 | 2.83 | 0.744 | 0.077 |
| DEADiff [54] | 0.295 | 10.81 | 1.14 | 2.66 | 0.721 | 0.089 |
| StyleDrop [62] | 0.292 | 12.16 | 1.28 | 2.51 | 0.717 | 0.090 |
| DreamStyler [3] | 0.301 | 11.90 | 1.09 | 2.53 | 0.691 | 0.095 |
| CSGO [74] | 0.298 | 6.43 | 0.83 | 3.09 | 0.716 | 0.089 |
| *SADis* (Ours) | **0.302** | **3.19** | **0.53** | **3.51** | 0.754 | 0.066 |

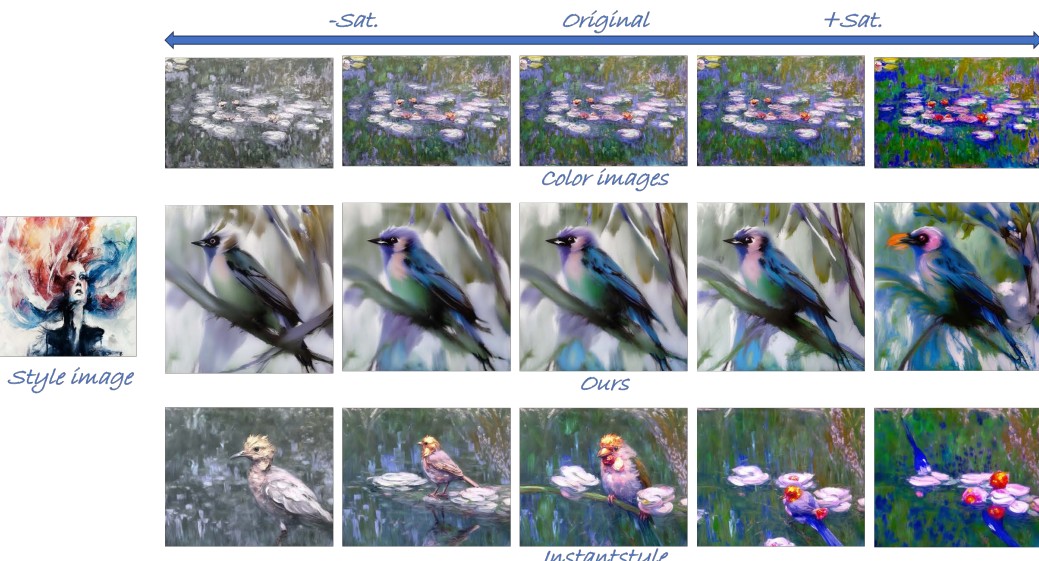

Figure 21: Compared to InstantStyle, *SADis* preserves the content more effectively under saturation variations.

## F    Ablation studies on *RegWCT*

***RegWCT* scales.** Given a latent $z_t'$ at timestep $t$, the color rectified latent $z_t$ after *RegWCT* is presented as:

$$z_t = (1 - \omega) \, z_t' + \omega \cdot \mathbf{RegWCT}(z_t'). \tag{7}$$

Here, we ablate the balancing factor $\omega$, as shown in Tab. 6. With the increasing scale $\omega$ of *RegWCT*, the color score of *SADis* improves significantly, while texture preservation remains largely unaffected. Based on the ablation results in Tab. 6, we set $\omega$ to 0.5 to achieve a balanced performance between color and texture alignment.

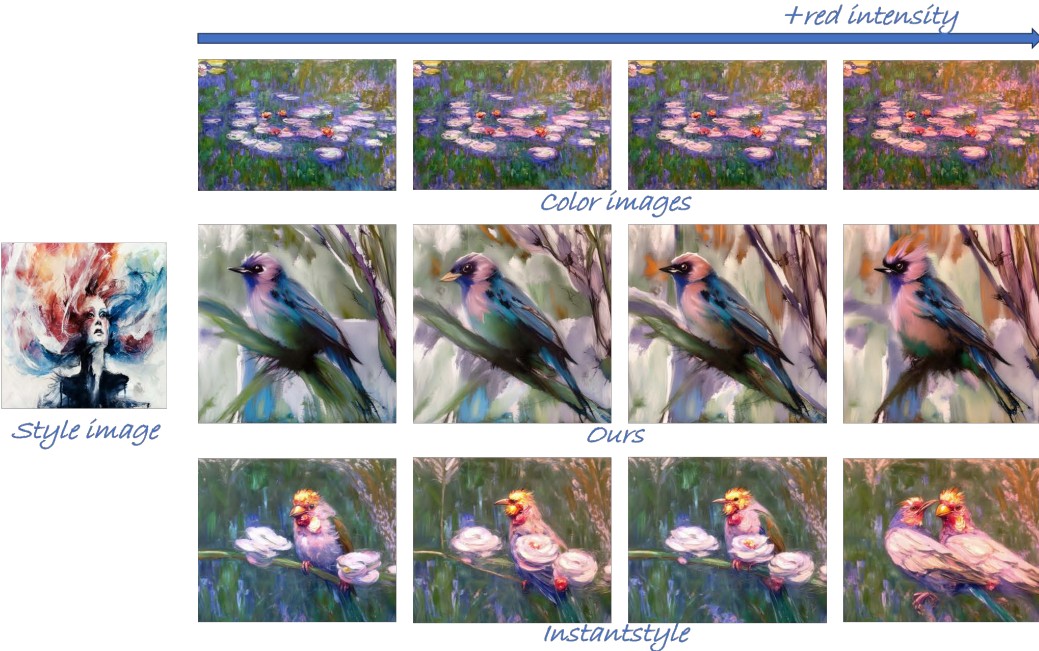

Figure 22: Compared to InstantStyle, *SADis* preserves the content more effectively under diverse illuminant variations.

Table 5: Quantitative results for different sampling rounds.

| Round | CLIP | Color | | | Texture | |
| --- | --- | --- | --- | --- | --- | --- |
| | | MS-SWD↓ | C-Hist↓ | GPT4o↑ | CLIP-I↑ | KID↓ |
| 1 | 0.301 | 6.08 | 0.95 | 3.14 | 0.751 | 0.064 |
| 2 | 0.301 | 6.05 | 0.95 | 3.15 | 0.750 | 0.065 |
| 3 | 0.300 | 6.23 | 0.96 | 3.12 | 0.752 | 0.064 |
| 4 | 0.301 | 6.16 | 0.95 | 3.12 | 0.751 | 0.066 |
| 5 | 0.304 | 6.08 | 0.95 | 3.14 | 0.749 | 0.066 |

**Timestep intervals.** The ablation study on different timesteps to apply *RegWCT* is shown in Fig. 23-(a). During the denoising process, applying *RegWCT* to the latent enhances the color alignment performance but sacrifices texture preservation. As observed in Fig. 23-(a), the early stages of denoising contribute more to the color than the latter stages, which is also supported by previous works [17, 81, 66]. Therefore, we only apply *RegWCT* during the early stages of the denoising, specifically within $[0.8T, 0.6T]$ to achieve a balance between the color and texture generation.

**Scale $\lambda$ of noise injection.** The ablation study on the noise scale $\lambda$ is illustrated in Fig. 23-(b). Without noise injection ($\lambda = 0$), applying WCT improves color alignment but also causes great texture degradation for *DisIG*. This issue is alleviated by injecting the specific degree of latent noise $z_T$, which is demonstrated in Fig. 23-(b). To achieve a balanced color and texture alignment, we set $\lambda$ to 0.01 in the experiments based on the results shown in Fig. 23-(b).

## G    Ablation studies on *CTE*.

**Scale $\gamma$ of SVD.** The ablation results of the scaling factor $\gamma$ and $\beta$ is presented in Tab. 7 and Tab. 8, respectively. As the scaling factor $\gamma$ increases, the color scores (such as MS-SWD and C-HIST) improve, with the sacrifice of the slightly texture performance degradation (revealed by CLIP-I). When $\beta$ increases beyond 1, the texture consistency metric (CLIP-I) improves, indicating enhanced texture alignment. However, this comes at the cost of decreased color consistency, as reflected by the MS-SWD and C-Hist scores. Therefore, to achieve a better balanced performance between texture and color consistency, we set the default values to $\beta = 1$ and $\gamma = 0.003$ in our main experiments.

Table 6: Ablation studies of *RegWCT* scales. considering the trade-off of texture and color alignment, the *RegWCT* scale is set as 0.5 by default.

| *RegWCT* Scale | | 0 | 0.3 | 0.5 | 0.7 | 1.0 |
|---|---|---|---|---|---|---|
| Color | MS-SWD (↓) | 8.05 | 5.65 | 5.57 | 5.23 | 5.18 |
| | C-Hist (↓) | 1.06 | 0.98 | 0.96 | 0.89 | 0.889 |
| Texture | CLIP-I (↑) | 0.747 | 0.744 | 0.743 | 0.741 | 0.740 |

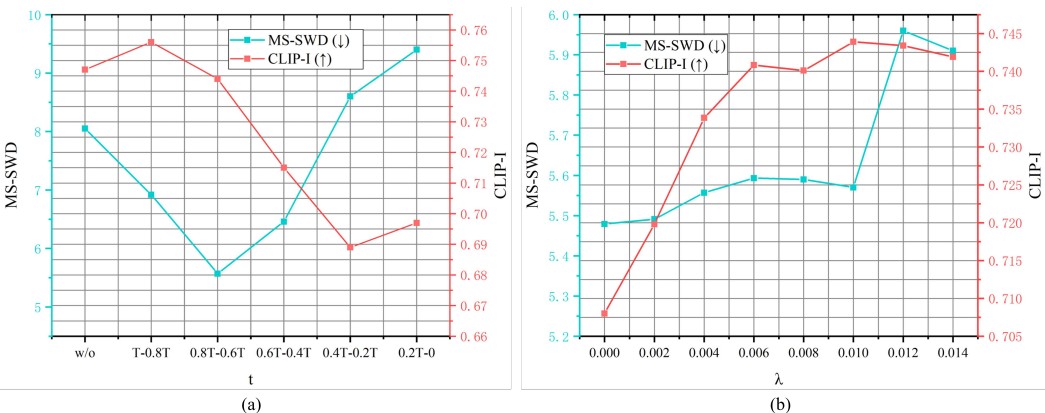

(a)       (b)

Figure 23: Ablation studies on applying *RegWCT* to different timestep intervals (left). Ablation studies on the scale $\lambda$ of noise injection (right) during applying *RegWCT*. By default, $\lambda$ is set to 0.01.

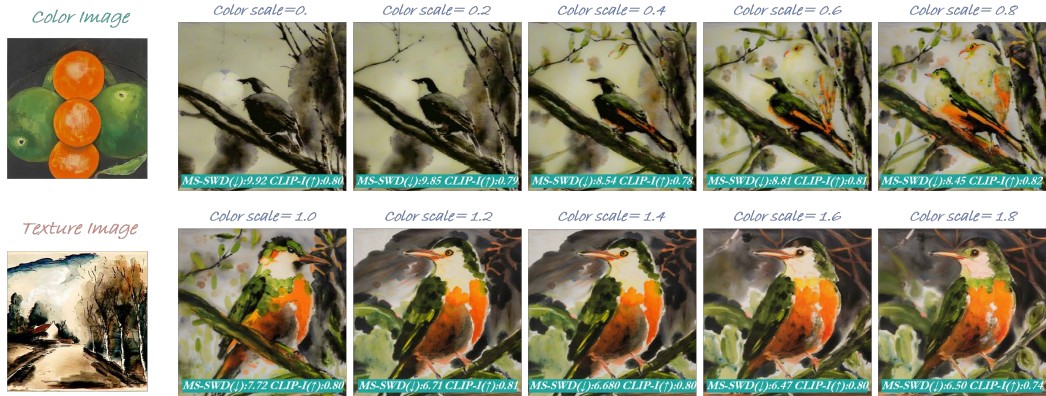

Figure 24: Results of different scales of color embedding $\mathbf{Emb}_{clr}$. Here, the scale of $\mathbf{Emb}_{tx}$ is fixed as 1.

Table 7: Ablation studies of scaling factor $\gamma$. $\gamma$ is set to 0.003 according to the ablation results.

| $\gamma$ ($\beta = 1$) | | 0 | 0.001 | 0.003 | 0.005 | 0.007 | 0.009 | 0.011 |
|---|---|---|---|---|---|---|---|---|
| Color | MS-SWD (↓) | 5.70 | 5.71 | 5.57 | 5.36 | 5.22 | 5.21 | 5.20 |
| | C-Hist (↓) | 1.01 | 1.017 | 0.962 | 0.939 | 0.897 | 0.887 | 0.861 |
| Texture | CLIP-I (↑) | 0.759 | 0.755 | 0.743 | 0.736 | 0.730 | 0.723 | 0.713 |

Table 8: Ablation studies of scaling factor $\beta$. $\beta$ is set to 1 according to the ablation results.

| $\beta$ ($\gamma = 0.003$) | | 0.5 | 0.7 | 0.9 | 1 | 1.1 | 1.3 | 1.5 |
|---|---|---|---|---|---|---|---|---|
| Color | MS-SWD (↓) | 5.20 | 5.28 | 5.45 | 5.57 | 5.70 | 5.92 | 6.17 |
| | C-Hist (↓) | 0.890 | 0.923 | 0.949 | 0.962 | 0.973 | 0.992 | 1.011 |
| Texture | CLIP-I (↑) | 0.711 | 0.729 | 0.740 | 0.743 | 0.747 | 0.749 | 0.752 |

Table 9: Ablation results of controlling different cross-attention layers of *SADis*.

| Different CA layers | CLIP | Color | | | Texture | |
| --- | --- | --- | --- | --- | --- | --- |
| | | MS-SWD↓ | C-Hist↓ | GPT4o↑ | CLIP-I↑ | KID↓ |
| b0a1(ours) | 0.301 | 6.08 | 0.95 | 3.14 | 0.751 | 0.064 |
| b0a0+b0a1 | 0.292 | 6.54 | 0.981 | 3.13 | 0.785 | 0.057 |
| b0a0+b0a1+b0a2 | 0.292 | 6.53 | 0.982 | 3.12 | 0.793 | 0.056 |
| b0a1+b1a0 | 0.291 | 6.55 | 0.981 | 3.11 | 0.791 | 0.057 |
| b0a1+b1a1 | 0.300 | 6.54 | 0.983 | 3.12 | 0.751 | 0.065 |
| b0a1+b1a2 | 0.297 | 7.19 | 1.005 | 3.10 | 0.753 | 0.066 |
| b0+b1 | 0.291 | 7.21 | 0.998 | 3.07 | 0.794 | 0.055 |
| b0+b1+b2 | 0.278 | 7.24 | 1.003 | 3.08 | 0.807 | 0.056 |

**Color-texture scales.** As shown in Fig. 24, we fix the scale of the texture embedding $\mathbf{Emb}_{tx}$ to 1 while varying the scale of the color embedding $\mathbf{Emb}_{clr}$ from 0 to 1.8. With a greater weight for the color embedding, the color score improves. However, it is worth noting that when the scale of the color embedding is set too high, it may negatively impact texture preservation and introduce artifacts that are not derived from either the color or texture reference images.

# H   Ablation studies on controlling different cross-attention layers.

The purpose of employing multiple Cross-Attention (CA) layers in IP-Adapter is to ensure consistency in the identity of the input. Prior researches [67, 68] indicate that different CA layers in IP-Adapter control different attributes. For example, some are more associated with content, while others relate more to style. Our decision to inject at the first decoder layer was initially inspired by InstantStyle [67]. Experiments with multiple CA layers are provided in Tab. 9. Here, 'bidx0aidx1' denotes the 'idx1'-th CA layer in the 'idx0'-th decoder block of the denoising UNet. As shown in the table, injecting into additional CA layers beyond the style-relevant one (i.e., the first decoder layer) improves texture metrics but degrades color representation. Moreover, we observed that controlling additional CA layers leads to generated images containing more semantic content from the texture reference image (similar to IP-Adapter's results shown in Fig. 1), which is inconsistent with the goal of texture transfer. This observation is consistent with the findings reported in InstantStyle.

