# OpenReview forum: "Free-Lunch Color-Texture Disentanglement for Stylized Image Generation"
_NeurIPS.cc/2025/Conference — NeurIPS 2025 poster_

### Official Review · Reviewer_3c1E · 2025-06-20

**Clarity:** 3
**Significance:** 3
**Originality:** 2
**Rating:** 4
**Confidence:** 3

**Summary:**

This paper proposes a novel approach called Style Attribute Disentanglement (SADis), designed to address a key challenge in stylized image generation: achieving independent control over color and texture. Specifically, SADis leverages the additive property of image prompts to disentangle color and texture for the first time without any training. This enables users to independently extract color and texture information from different reference images.

**Questions:**

My major concerns are the weakness. I would like the authors to directly address these concerns in their response. My final rating is subject to change based on the quality and clarity of the authors’ feedback.

**Ethical Concerns:**

["NO or VERY MINOR ethics concerns only"]

**Final Justification:**

The proposed method, SADis, built upon the U-Net architecture, demonstrates a simple yet effective approach for controlling color and texture. However, the exploration of controllable generation within more recent and mainstream architectures such as MMDiT remains an open and promising direction. I look forward to the authors’ future work addressing this aspect.

**Limitations:**

The authors are encouraged to answer the questions and address the weakness above.

**Paper Formatting Concerns:**

No format issues.

**Quality:**

3

**Strengths And Weaknesses:**

Strengths

1、The method is easy to follow. This paper proposes using the CLIP model to extract disentangled color and texture features from reference images and guide the generation of target images through the IPA plugin. Both qualitative analysis and quantitative metrics demonstrate strong performance.

2、It requires no additional data training, can leverage any existing T2I model, and is fully compatible with the SD community ecosystem.

Weaknesses

1、The ablation study in Table 2 suggests that the proposed component contributes primarily to the injection of style color information, with limited impact on texture preservation. This raises the question of whether the texture-related performance gain is mainly attributable to the IPA module, rather than the disentangled texture features themselves.

2、The original IPA module applies cross-attention across multiple decoder layers, whereas the proposed method restricts injection to only the first decoder layer. It remains unclear whether this simplification affects the integrity of the generation process, potentially limiting the model’s ability to integrate style features progressively.

3、The ablation study does not sufficiently isolate the effect of the proposed TextureExtraction module. It is therefore difficult to determine whether improvements in texture retention are due to this module or the inherent capabilities of IPA. More targeted experiments would help clarify this.

4、How significant is the impact of the singular value modulation parameters γ (gamma) and β (beta) on the generation process? How are they set during training, and are they learnable.

---

> ### Author Rebuttal · Authors · 2025-07-31
>
> We sincerely appreciate your constructive and insightful feedback. We are grateful for your recognition that the method is ***compatiblewith the community ecosystem***  and the ***performance is strong***. Below, we provide detailed responses to each of your specific points.
>
>
>
>
> **W1(texture-related performance gain)**: In our method, the ability to represent textual information primarily comes from the pre-trained model (i.e., IPA and SD). However, IPA injects image features into all blocks of the network, which introduces semantic leakage from the reference image and results in poor text-image alignment. In contrast, InstantStyle only injects features into style-relevant blocks, effectively avoiding the semantic leakage issue inherent in the original IPA.
> We follow the InstantStyle injection strategy, which enables better adherence to textual semantics. As a result, the texture-related performance of our method aligns with that of InstantStyle, as shown by the CLIP-I scores in Table 1.
>
>
>
>
> **W2(supplement experiments with multiple CA layers)**: The purpose of employing multiple Cross-Attention (CA) layers in IPA is to ensure consistency in the identity of the input. Prior research has shown that different CA layers in IPA control different attributes—for example, some are more associated with content, while others relate more to style. Our decision to inject at the first decoder layer was initially inspired by InstantStyle.
> Experiments with multiple CA layers are provided below. Here, ‘b{idx0}a{idx1}’ denotes the ‘idx1’-th CA layer in the ‘idx0’-th decoder block of the denoising UNet. As shown in the table,  injecting into additional CA layers beyond the style-relevant one (i.e., the first decoder layer) negatively affects color representation, despite improvements in texture metrics. Moreover, from the visualization results, we observed that the generated images tend to contain more semantic content from the texture reference image (similar to IPA’s results shown in Fig. 5), which is inconsistent with the goal of texture transfer. This phenomenon is consistent with the findings in InstantStyle.
>
>
>
> |     Different CA layers    |     CLIP↑    |     Color (MS-SWD↓)    |     Color (C-Hist↓)    |     Color (GPT4o↑)    |     Texture (CLIP-I↑)    |     Texture (KID↓)    |
> |:--------------------------:|:------------:|:----------------------:|:----------------------:|:---------------------:|:------------------------:|:---------------------:|
> |          b0a1(ours)        |     0.301    |         6.08           |         0.95           |        3.14           |         0.751            |        0.064          |
> |          b0a0+b0a1         |     0.292    |         6.54           |        0.981           |        3.13           |         0.785            |        0.057          |
> |        b0a0+b0a1+b0a2      |     0.292    |         6.53           |        0.982           |        3.12           |         0.793            |        0.056          |
> |          b0a1+b1a0         |     0.291    |         6.55           |        0.981           |        3.11           |         0.791            |        0.057          |
> |          b0a1+b1a1         |     0.300    |         6.54           |         0.983          |        3.12           |         0.751            |        0.065          |
> |          b0a1+b1a2         |     0.297    |         7.19           |        1.005           |        3.10           |         0.753            |        0.066          |
> |            b0+b1           |     0.291    |         7.21           |        0.998           |        3.07           |         0.794            |        0.055          |
> |           b0+b1+b2         |     0.278    |         7.24           |        1.003           |        3.08           |         0.807            |        0.056          |
>
> When texture and color embeddings are injected solely into the style-related CA layer (i.e., the first decoder layer), the visual results show more desirable style transfer performance. We will include the corresponding visualization results in the supplementary material.
>
>
>
>
> **W3(clarification of TextureExtraction module)**: We apologize for any confusion. In our paper, the “SVD” mentioned in Table 2 and in the ablation experiments of Fig. 4 actually refers to the TextureExtraction module. We will revise the terminology accordingly in future versions.
>
>
> **W4(Ablation on γ and β)**: We conduct an ablation study on γ and β. As shown in the table below, when β increases beyond 1, the texture consistency metric (CLIP-I) improves and even surpasses that of InstantStyle (CLIP-I = 0.74), indicating enhanced texture alignment. However, this comes at the cost of decreased color consistency, as reflected by the MS-SWD and C-Hist scores. Therefore, to achieve a better balanced performance between texture and color consistency, we set the default values to β = 1 and γ = 0.003 in our main experiments.
>
> |             γ   (β=1)           |       0      |     0.001    |     0.003    |     0.005    |     0.007    |     0.009    |     0.011    |
> |:-------------------------------:|:------------:|:------------:|:------------:|:------------:|:------------:|:------------:|:------------:|
> |      Color     (MS-SWD   ↓)     |      5.70    |      5.71    |      5.57    |      5.36    |      5.22    |      5.21    |      5.20    |
> |      Color     (C-Hist   ↓)     |      1.01    |     1.017    |     0.962    |     0.939    |     0.897    |     0.887    |     0.861    |
> |     Texture     (CLIP-I   ↑)    |     0.759    |     0.755    |     0.743    |     0.736    |     0.730    |     0.723    |     0.713    |
>
>
> |           β (γ=0.003)         |     β=0.5    |     β=0.7    |     β=0.9    |      β=1     |      β=1.1    |     β=1.3    |     β=1.5    |
> |:-----------------------------:|:------------:|:------------:|:------------:|:------------:|:-------------:|:------------:|:------------:|
> |     Color     (MS-SWD   ↓)    |      5.20    |      5.28    |      5.45    |      5.57    |      5.70     |      5.92    |      6.17    |
> |     Color     (C-Hist   ↓)    |     0.890    |     0.923    |     0.949    |     0.962    |      0.973    |     0.992    |     1.011    |
> |      Texture    (CLIP-I ↑)    |     0.711    |     0.729    |     0.740    |     0.743    |      0.747    |     0.749    |     0.752    |

---

> > ### Comment · Reviewer_3c1E · 2025-08-06
> >
> > Thank you for the authors’ detailed response. The experiments conducted have effectively addressed my primary concerns. As SADis is currently implemented as a conditional control mechanism within the U-Net architecture, I am curious about how similar designs could be extended to models based on alternative structures such as MMDiT and FLUX. I look forward to the authors’ future work in this direction. Based on the clarifications and improvements, I am raising my rating.

---

> > > ### Author Response · Authors · 2025-08-07
> > >
> > > Dear Reviewer 3c1E,
> > >
> > > Thank you very much for your suggestions. We conduct further experiments and find that applying SADis to two MMDiT-based models (SD3.5 and FLUX.1-dev) can also achieve color-texture disentangled stylization.
> > >
> > > Specifically, we use SD3.5 and FLUX.1-dev as the base models. These models employ the original text encoders (i.e., T5, CLIP-L, and CLIP-G) for content control, and utilize SigLIP with a pretrained adapter for style control. We further use the features from SigLIP to extract and disentangle color-texture embeddings, consistent with the SADis workflow.
> > >
> > > Experimental results show that SADis can achieve color-texture disentanglement and enable stylized image generation on these MMDiT-based models as well. This demonstrates that **the color-texture disentanglement capability of SADis does not rely on the U-Net architecture and can be generalized to MMDiT-based models** for color-texture disentanglement in stylized image generation. We will include the corresponding results in our final version.
> > >
> > > Thanks again for your valuable suggestions.
> > >
> > > Best regards,
> > >
> > > The Authors

---

### Official Review · Reviewer_Z8Hi · 2025-06-26

**Clarity:** 3
**Significance:** 4
**Originality:** 4
**Rating:** 5
**Confidence:** 4

**Summary:**

This paper investigates the additivity of the CLIP image embedding, especially for extracting color information of arbitrary images. Based on the observation, the authors propose a novel method, namely SADis, that disentangles the learning of color and texture to represent style characteristics. To remove the redundant color information from the texture image, several strategies are introduced, manipulating image embeddings by average operation, SVD, and WCT. Experiments show the disentanglement of color and texture benefits stylized image generation, as well as in-depth investigations on a variety of possible applications.

**Questions:**

1. The setting of "Baseline" in Fig. 4 (a) is not clear. I found it in lines 175-177 as "using only the grayscale texture embedding". Does this mean the texture branch w/o averaged embedding, SVD, and RegWCT?

2. Since previous methods lack disentanglement ability, why not use the same style image for both the color and texture references to SADis, and then compare with these methods? In my opinion this would be fairer than using generated captions. In this case, I am curious about the difference between SADis, IP-Adapter, and InstantStyle, as the other two methods seem to capture texture very well but are inferior in color (observe from Fig. 5 and  user study in Tab. 1)

3. Ablation studies are only conducted on two components, i.e., SVD and RegWCT. I found these two components contribute more on the color consistency in Tab. 2. It seems that they help remove color information from the texture image to inject color information, but I guess the learning of texture might benefit from another design, e.g., the concatenated averaged embedding, in which an ablation study is missing.

Minor suggestions:
1. Fig.4 better provides the text prompts or the content images (if content-conditioned generation using ControlNet) for the generated images.

2. Should the line 177 be "Fig. 4 (a)-2nd column", line 193 be "Fig. 4 (a)-3rd column", and line 226 be "Fig. 4 (a)-4th column"?

3. Fig. 4 (b) the description "+z_T injection", if it is the formulation in line 217, I recommend rephrasing it into noise injection, or directly RegWCT, as T is the starting point, but this injection seems to perform in a period.

**Ethical Concerns:**

["NO or VERY MINOR ethics concerns only"]

**Final Justification:**

This paper investigates an interesting phenomenon--the additivity property of the pre-trained CLIP image embedding space. Motivated by this, the authors propose the SADis method that disentangles color and texture attributes from style images. While SADis builds on  IP-Adapter for texture extraction, it shows notable improvement particularly in color and texture disentanglement, as evidenced in Fig. 5 and quantitative comparisons. This work provides valuable insights into understanding the color representation and has great potential to inspire future research. Therefore, I recommend a score of 5 with a confidence of 4.

**Limitations:**

The authors describe the limitations of the proposed SADis method in the appendix.

**Quality:**

3

**Strengths And Weaknesses:**

**Strengths**:

The paper is well-written and easy to follow. The phenomenon of image-prompt additivity is very intuitive, and additional analysis in the Appendix is also interesting. I really like these illustrations, which help me to grasp the idea quickly. The final method is well motivated based on the observations. The evaluation is thorough and demonstrates the effectiveness of the proposed method very well. I also appreciate the human judgment and time cost reported.

**Weaknesses**:

While the method clearly surpasses existing style transfer methods in the color-texture-content reference scenario, the authors use LLM-generated descriptions during comparison. This setting might be unfair for other approaches.

---

> ### Author Rebuttal · Authors · 2025-07-31
>
> We sincerely appreciate your constructive and insightful feedback. We are grateful for your recognition that the method is ***intuitive*** and the analysis is ***interesting***, the design is ***well-motivated***, and the paper is with ***good writing***. Below, we provide detailed responses to each of your specific points.
>
>
>
> **Q1(clarification of the baseline)**: Your understanding is absolutely correct. We apologize for any inconvenience caused by the lack of clarity. In Fig.4(a), Our ‘Baseline’ denotes color-texture disentanglement with Image-Prompt Additivity only (i.e., without SVD and RegWCT). In the revised version, we will add clarifications regarding our ‘Baseline’ in the captions.
>
>
>
>
> **Q2&W1(experiments with the same reference image)**: According to your suggestion, we have added results where both texture and color are derived from the same reference image, using the same prompt as in R1.Q2. The comparative experimental results are presented in the table below. In any future versions, we plan to include these results in the supplementary materials. As shown by the experimental results, our method still achieves the best color consistency, which is consistent with the findings reported in the original paper.
>
> In terms of texture consistency, our method achieves the best performance among all methods except for IP-Adapter. However, it is important to note that although the IP-Adapter achieves the highest numerical score for texture consistency, it introduces severe semantic leakage from the texture reference image (see Row 2 and Row 3 in Fig. 5), resulting in poor text-image alignment (CLIP: 0.260 in the table below), which fails to meet the requirements of stylization. In contrast, excluding the IP-Adapter, our method achieves the highest scores in color consistency (MS-SWD: 3.19), texture consistency (CLIP-I: 0.754), and text-image alignment (CLIP: 0.302).
>
>
> |        Method       |     CLIP↑    |     Color (MS-SWD↓)    |     Color (C-Hist↓)    |     Color (GPT4o↑)    |     Texture (CLIP-I↑)    |     Texture (KID↓)    |
> |:-------------------:|:------------:|:----------------------:|:----------------------:|:---------------------:|:------------------------:|:---------------------:|
> |         SDXL        |     0.292    |         9.12           |         1.15           |        3.04           |         0.696            |        0.090          |
> |       IPAdapter     |     0.260    |         4.18           |         0.63           |        3.44           |         0.815            |        0.057          |
> |     InstantStyle    |     0.277    |         4.21           |         0.64           |        3.41           |         0.753            |        0.066          |
> |        Artist       |     0.272    |        10.98           |         1.18           |        2.83           |         0.744            |        0.077          |
> |        DEADiff      |     0.295    |        10.81           |         1.14           |        2.66           |         0.721            |        0.089          |
> |       StyleDrop     |     0.292    |        12.16           |         1.28           |        2.51           |         0.717            |        0.090          |
> |      dreamstyler    |     0.301    |        11.90           |         1.09           |        2.53           |         0.691            |        0.095          |
> |         CSGO        |     0.298    |        6.43           |          0.83           |        3.09           |         0.716            |        0.089          |
> |     SADis (Ours)    |     0.302    |        3.19           |          0.53           |        3.51           |         0.754            |        0.066          |
>
>
> In addition, it should be noted that the ability of our method to represent textual information primarily originates from the pre-trained model (i.e., IP-Adapter and InstantStyle). However, IP-Adapter injects image features into all blocks of the network, which leads to semantic leakage  (see Row 2 and Row 3 in Fig. 5) from the reference image and results in poor text-image alignment . In contrast, InstantStyle only injects features into style-relevant blocks, effectively avoiding the semantic leakage issue present in the original IP-Adapter. We adopt the same injection strategy as InstantStyle, which allows for better adherence to the textual description rather than the semantics of the reference image. Given that our method and InstantStyle utilize the same style injection strategy, it is reasonable that their texture performance is comparable. Nevertheless, our main contribution lies in further improving color consistency while maintaining texture performance on par with InstantStyle, thereby achieving effective color-texture disentanglement.
>
>
>
>
>
> **Q3(ablation study of concatenation)**:
> Regarding the concatenated averaged embedding, it serves as the basis for SVD-based grayscale suppression, and therefore, SVD cannot be separated from the concatenation operation. The purpose of this concatenation is to enable the identification of the principal singular value(s) corresponding to grayscale information in the feature space after the SVD transformation.
>
> Your understanding regarding the improvements in color consistency brought by SVD and RefWCT is correct. To achieve stylized image generation, we extract color information from the image-prompt and texture information from real images. When isolating color from the image prompt, two key challenges arise: (1) extracting color while minimizing the influence of texture from the image-prompt, and (2) removing color information from the real image to prevent it from interfering with the intended color guidance.
> The first challenge can be easily addressed by our proposed color-extraction method. To address the second challenge, we adopt SVD to suppress the color influence of the real image, enabling the extraction of cleaner texture features (as shown in Figure 2, Texture Extraction). Although SVD effectively reduces the impact of real-image colors on the final generation, it leads to only a slight degradation in texture fidelity, evidenced by a minor 0.01 drop in the CLIP-I score.
> Furthermore, your understanding of the role of the concatenated averaged embedding is also accurate. This embedding serves as the foundation for the SVD-based grayscale suppression. Therefore, SVD cannot be decoupled from the concatenation operation. The purpose of this concatenation is to identify the principal singular value(s) associated with grayscale information in the feature space after the SVD transformation.
>
> To ensure better color consistency, we further introduce RegWCT, which enhances the traditional Whitening and Coloring Transform (WCT) by incorporating noise regularization in the latent space of diffusion models. This modification further improves color fidelity, again with only a minimal impact on image quality (a 0.01 drop in CLIP-I).
>
>
> **Minor points**: Thank you very much for your suggestion. We will incorporate your feedback in our final revisions.

---

> > ### Comment · Reviewer_Z8Hi · 2025-08-03
> > **Response to rebuttal**
> >
> > Thanks for the rebuttal. I appreciate the authors' engagement with all reviewers' concerns. From the table in the response to Q2, SADis shows improvement over IP-Adapter and InstantStyle in terms of color, albeit with inferior texture (which I guess is partly due to the two metrics used for texture are not suitable, while texture is difficult to quantify/recognize even for humans). These results are consistent with the authors' statement. Additionally, could the authors also provide the time cost of IP-Adapter and InstantStyle? As Table 2 only reports that of SADis, such a comparison would better assess SADis and the other two works.

---

> > > ### Author Response · Authors · 2025-08-04
> > > **Inference time of IP-Adapter and InstantStyle**
> > >
> > > Thank you for your feedback and for your appreciation of our engagement.
> > >
> > > Regarding the time cost of IP-Adapter and InstantStyle, we have measured their inference times under the same hardware and settings as SADis. The results are as follows:
> > >
> > > **IP-Adapter: 9.52 seconds per image**
> > >
> > > **InstantStyle: 9.31 seconds per image**
> > >
> > > We will include these results in the future revised version of Table 2 and discuss their implications in the main text to provide a more balanced assessment.

---

> > > > ### Comment · Reviewer_Z8Hi · 2025-08-06
> > > >
> > > > Thanks for reporting the inference time of the two methods! Including these results in the final version is indeed beneficial. The authors' response has addressed all my concerns. I will maintain my current rating and confidence.

---

> > > > > ### Author Response · Authors · 2025-08-06
> > > > >
> > > > > Thank you very much for your positive feedback. We sincerely appreciate your valuable comments and thoughtful suggestions throughout the review process, which have been very helpful in improving our work.

---

### Official Review · Reviewer_N4y7 · 2025-07-01

**Clarity:** 3
**Significance:** 3
**Originality:** 3
**Rating:** 4
**Confidence:** 3

**Summary:**

This paper introduces a tuning-free method of disentangling color and texture for stylized image generation. The authors discover that the color embedding and textural embedding can be added as the style condition. The authors propose some techniques to improve generation quality, such as SVD embedding optimization and regularized whitening-coloring transforms.

**Questions:**

Please see weaknesses.

**Ethical Concerns:**

["NO or VERY MINOR ethics concerns only"]

**Final Justification:**

The authors address my concerns, especially by providing experiments of SADis combined with other vision encoders (DINOv2 and SigLIP). I would like to keep my original rating and lean toward acceptance.

**Limitations:**

Yes.

**Paper Formatting Concerns:**

No.

**Quality:**

3

**Strengths And Weaknesses:**

**Strengths**

- The idea of disentangling color and texture information addresses a practical need.
- The proposed method is tuning-free, making it easily applicable to various backbones such as IP-Adapter and ControlNet.
- The design choice of using additive CLIP embeddings, along with further optimizations via SVD or WCT, is reasonable.
- The generated results are of good quality.

**Weaknesses**

- From Table 2, SVD improves color persistence but worsens texture, especially when combined with RegWCT.
- The authors propose various techniques, such as SVD, WCT, and noise regularization. However, all of these methods aim to strike a better trade-off between color and texture quality, and no single component is able to improve both aspects simultaneously.
- This method relies on the additive property of CLIP as a vision encoder, but its effectiveness with other encoders, such as T5 or DINO, has not been validated.

---

> ### Author Rebuttal · Authors · 2025-07-31
>
> We sincerely appreciate your constructive and insightful feedback. We are grateful for your recognition that the method is ***applicable*** and ***addresses practical needs***, the design is ***reasonable***, and the generation is with ***good quality***. Below, we provide detailed responses to each of your specific points.
>
>
> **W1&W2(texture consistency)**:
> The objective of our method is to explore tuning-free color-texture disentanglement using pre-trained models. ***In our approach, texture consistency is primarily inherited from the pre-trained models(InstantStyle), while our goal is to enhance color consistency with minimal loss of texture***, thereby achieving a balance between color and texture attributes. To this end, both SVD and RegWCT are designed to address poor color consistency while minimizing texture loss as much as possible.
>
> In terms of texture performance as reported in Table 1, our method achieves texture consistency second only to IP-Adapter and is comparable to InstantStyle (Our base model). However, it is important to note that although the IP-Adapter achieves the highest numerical score for texture consistency, it introduces severe semantic leakage from the texture reference image (see Row 2 and Row 3 in Fig. 5), resulting in poor text-image alignment (CLIP: 0.233), which fails to meet the requirements of stylization. In contrast, apart from the IP-Adapter, our method achieves the highest scores in color consistency (MS-SWD: 5.57), texture consistency (CLIP-I: 0.74), and text-image alignment (CLIP: 0.281). These results are also supported by the user study in Table 2 and Fig. 4. Additionally, following the suggestion of Reviewer R1.Q2, we have included performance comparisons under complex prompts, and the resulting trends are consistent with the above analysis. Overall, our method focuses on improving color consistency while maintaining texture consistency, and the experimental results demonstrate that our approach achieves a better balance between color and texture consistency.
>
> We agree that our method shows stronger improvements in color accuracy, with slightly lower texture scores. However, it is important to note that the texture metrics used (e.g., CLIP-I, KID) remain sensitive to color content: modifying the color palette inevitably affects the greyscale values, which play a role in defining textures. Consequently, methods that transfer both color and texture from the same reference image—effectively ignoring the color image—benefit from aligned visual cues, making it easier to achieve higher texture scores. In contrast, our method explicitly disentangles color and texture sources and faithfully adapts to the color reference image. This necessary shift in greyscale structure may lead to a modest drop in texture metrics.
>
>
>
> **W3(other encoders)**: Our method is training-free and built upon IP-Adapter. Since IP-Adapter relies solely on the CLIP encoder as its vision encoder and does not utilize other encoders such as T5 or DINO from pre-trained models, conducting such an evaluation is not feasible. Furthermore, our method can be extended to any backbones as long as there is an IP-Adapter-like pretrained adapter for the backbone model, including Flux, SD3, SD3.5, etc.

---

> ### Comment · Reviewer_N4y7 · 2025-08-04
>
> Thank you for the rebuttal. Regarding W3, I understand the work is built on IP-Adapter and cannot be easily validated with other encoders. The authors could consider demonstrating the additive property of other encoders (e.g., DINO or T5) in the final version, to better support the feasibility of integrating diverse encoders into alternative architectures. Overall, I will maintain my current rating.

---

> > ### Author Response · Authors · 2025-08-06
> >
> > Thank you so much for your feedback and valuable suggestions. Considering that T5 is a text encoder and is commonly used to encode text prompts instead of image prompts in text-to-image models (e.g., SD3, FLUX), it is not feasible to verify image embedding additivity with T5. To evaluate the applicability of SADis with other image encoders, we are currently experimenting with the feasibility of using the DINOv2 image encoder. For training the IPA-Adapter for DINOv2, we need training a lightweight adapter from scrach to achieve image-controlled generation capability for the pre-trained text-to-image diffusion models. Thus, it requires a certain amount of time. We will provide further feedback and update the supplementary materials if results become available.

---

> > ### Author Response · Authors · 2025-08-08
> >
> > Dear Reviewer N4y7,
> >
> > Thank you very much for your thoughtful and insightful suggestions, which are helpful for improving our paper.
> >
> > We have conducted further experiments and found that **combining SADis with DINOv2 can also achieve color-texture disentangled stylization**.
> >
> > Specifically, we use DINOv2-large as the image encoder and the photo-concept-bucket dataset (about 560K text-image pairs). Following the IP-Adapter training strategy, we trained an adaptor for DINOv2-large from scratch. So far, we have completed 170K training iterations and observed that the model has started to produce promising results. We evaluated the latest checkpoint and found that, following the SADis pipeline, DINOv2-large with the adaptor is able to achieve color-texture disentanglement for stylization.
> >
> > However, there are still some limitations: the visual quality of the color-texture disentangled images generated using DINOv2-large is slightly inferior to those generated using CLIP. We speculate that this is partly due to the relatively fewer training iterations (170K vs. 1M iterations for IP-Adapter) and the smaller dataset (560K vs. 10M text-image pairs for IP-Adapter). More importantly, compared to CLIP, the features extracted by DINOv2-large are not aligned with text, and instead capture more semantic information from images. This makes the compositionality more semantic-oriented, and thus there is a higher risk of semantic leakage from the reference image in the generated results.
> >
> > In addition, we also experiment with two pre-trained MMDiT-based models (SD3.5 and FLUX.1-dev) using **SigLIP** as the image encoder. We find that applying SADis to these models can also achieve effective color-texture disentangled stylization.
> >
> > These findings suggest that **SADis can be extended to other image encoders (such as CLIP, DINOv2, and SigLIP) and base models with different architectures (e.g., SDXL, SD3.5, and FLUX.1-dev)**. We will include these experimental results and related findings in the final version of our paper.
> >
> > We hope the above findings and analysis address your concerns.
> >
> > Best regards,
> >
> > The Authors

---

### Official Review · Reviewer_L5Gk · 2025-07-03

**Clarity:** 3
**Significance:** 2
**Originality:** 2
**Rating:** 4
**Confidence:** 4

**Summary:**

The author proposes a tuning-free style-preserving scheme for texture and color disentanglement. Leveraging the additive property of the CLIP embedding space and assuming that image style features can be decomposed into color and texture features as a strong hypothesis, the disentanglement scheme is designed. Different from existing methods that require model fine-tuning, SADis achieves style disentanglement solely through forward propagation and feature transformation, without modifying the parameters of pre-trained diffusion models, thus significantly reducing computational costs.

**Questions:**

1.How to verify the correctness of the strong hypothesis in the process of Eq1?

2.The author demonstrates some simple prompt scenarios, but lacks verification with complex prompts.

3.How to verify the correctness of the strong hypothesis in the process of Eq3?

4.I'm more concerned about the issue of stability. Will there be significant differences when generating multiple images (rolling).

5.Some recent methods only support single-image processing. How to perform inference for multiple images?

**Ethical Concerns:**

["NO or VERY MINOR ethics concerns only"]

**Final Justification:**

The rebuttal provided helpful clarifications and new experimental results that addressed my concerns. I recommend including these analyses in the main paper to improve clarity and completeness. Thus, I am increasing my score.

**Paper Formatting Concerns:**

There is a grammatical error in line 71.

There seems to be a logical error in lines 61-62.

**Quality:**

2

**Strengths And Weaknesses:**

Strength:

1. The writing of the paper is clear.

2.The motivation of the paper is clear.

3. The method proposed in the paper is interesting, which decouples CLIP features to obtain texture and color information. However, I still worry that such information is unstable.

4. The paper provides some quantitative and user studies.

Weaknesses:

1.Many contents of the paper are based on some strong assumptions, which may lack sufficient theoretical basis.

2.Lack of verification with some complex prompts.

3.The visualization results in Figure 5 do not demonstrate that the proposed method has significant advantages.

---

> ### Author Rebuttal · Authors · 2025-07-31
>
> We sincerely appreciate your feedback. We are grateful for your recognition that the paper is ***well-written***, the research motivation is ***clear***, and our proposed method is ***interesting***. Below, we provide detailed responses to each of your specific points.
>
> **W1&Q1(strong hypothesis in Eq.1):** Our verification of the image-prompt additivity hypothesis in Eq. 1 is empirical rather than formally proven, and this approach is consistent with established practices in analyzing learned representations of T2I models[1,2]. Prior works such as SEGA [3] and ToMe [4] have demonstrated that CLIP-encoded embeddings exhibit semantic compositionality through additive and subtractive properties in textual spaces, providing foundation for similar behavior in image embedding spaces. Furthermore, the difference vectors between pairs of images with the same color (e.g., A − A* and B − B*) point in consistent directions (Fig. 3a), demonstrating the correctness of this subtraction-based color decoupling method. Besides, the consistent superior performance in color alignment metrics (MS-SWD, C-Hist, GPT-4 color scores) and successful disentanglement results shown in Fig. 4-Fig.6 also provide strong empirical evidence for the correctness of this hypothesis.
>
> **W1&Q3(correctness of Eq.3)**: Eq. 3 is inspired by WNNM[5], a method originally proposed for image denoising. WNNM assumes that most of the noise in a noisy image is concentrated in the lower-K singular values. Each singular value σ of an image patch is updated according to the formula:
> $$\sigma \leftarrow \sigma - \frac{\lambda}{\sigma + \varepsilon}$$
>
> In this paper, we apply **Eq. 3** to further suppress the gray color in grayscale images, while preserving only the texture information. Specifically, our goal is to decouple texture from the *textural image* and eliminate the influence of color information. We observe that directly using the Y channel (grayscale) of the textural image introduces a grayish color bias in the generated results (see Fig. 4(a), *Baseline*). To address this, we aim to remove the residual gray information from the grayscale image.
>
> To achieve this, we concatenate the Y channel of the textural image with its mean grayscale intensity (i.e., the average gray value), forming a new embedding using **Eq. 2**. This embedding mainly captures the gray color information. Therefore, the **top-K singular values** in the constructed embedding matrix $\text{Emb}'_{\text{tx}}$ predominantly correspond to the color information we intend to suppress. At the same time, WNNM demonstrates that singular values carry clear physical meaning and should be weighted differently. Thus, we adopt **Eq. 3** to ensure that components corresponding to **larger singular values** undergo **stronger shrinkage**, effectively removing the undesired gray color while retaining informative texture features.
>
> **W2&Q2 (complex prompts):**
> To further evaluate performance with complex prompts, we conducted additional experiments by randomly sampling 10 complex prompts from DREAMBENCH++ [6], together with 10 color and 10 texture images from our dataset, resulting in 1,000 generated images for evaluation.
> The performance trend under long and complex prompts remains consistent with the results in our main paper (Table 1). Compared to other methods, our approach achieves the best disentanglement of color and texture, resulting in a more balanced performance in both color and texture consistency. Specifically, our method attains significantly higher color consistency, while other methods often exhibit strong color-texture entanglement, with color being overly influenced by the texture reference image.
> Although IP-Adapter achieves the highest numerical score for texture consistency, it suffers from severe semantic leakage from the texture reference image (see Row 2 and Row 3 in Fig. 5), leading to poor text-image alignment (CLIP score: 0.257), which is inadequate for stylization. In contrast, except for IP-Adapter, our method achieves the highest scores in both texture consistency and text-image alignment.
>
> |Method|CLIP↑|Color(MS-SWD↓)|Color(C-Hist↓)|Color(GPT4o↑)| Texture(CLIP-I↑)| Texture(KID↓)|
> |:-:|:-:|:-:|:-:|:-:|:-:|:-:|
> |SDXL|0.291|9.00|1.14|3.07|0.698|0.091|
> |IP-Adapter|0.257|9.99|1.15|2.96|0.817|0.058|
> |InstantStyle|0.278|11.21|1.25|2.85|0.751|0.065|
> |Artist|0.253|11.15|1.24|2.67|0.742|0.082|
> |DEADiff|0.280|10.01|1.14|2.80|0.718|0.090|
> |StyleDrop|0.290|11.90|1.35|2.83|0.731|0.078|
> |Dreamstyler|0.300|11.27|1.21|2.57|0.677|0.083|
> |CSGO|0.299|12.60|1.22|2.64|0.728|0.081|
> |SADis(Ours)|0.301|6.08|0.95|3.14|0.751|0.064|
>
> **W3 (visual results in Fig. 5):** Compared to the comparison methods, our approach offers best ***disentanglement of color and texture***, achieving a more balanced performance in terms of color and texture consistency.
> As shown in Fig. 5, even with prompts providing detailed color descriptions from the reference image, baseline methods fail to accurately reproduce the intended colors. In contrast, our method achieves much higher color consistency and avoids the color-texture entanglement seen in other approaches, where color is overly influenced by the texture reference image. Notably, our method maintains strong texture preservation while improving color consistency. For instance, in Fig. 4 (last row), our generated "bear" image better matches the color of the “wave” reference, while other methods—such as StyleDrop, IP-Adapter, InstantStyle, DreamStyler, DEADiff, and CSGO—tend to inherit the colors of Van Gogh's Starry Night (the texture image). Although SDXL and Artist also produce colors closer to the “wave” image, their results are overly coarse. Similar trends appear in examples in Fig. 4, with additional results on color-texture disentanglement provided in Fig. S6 of the supplementary material.
>
> **Q4(stability of generating multiple images)**:  To assess the stability of color and texture in repeated generations, we conducted additional experiments by using the data in **W2&Q2**. For each setting, the process was repeated five times. In each round, 1,000 images were generated, and the relevant metrics were computed. The results, summarized in the table below, show minimal variation in color and texture consistency across rounds, demonstrating the robustness and stability of our method.
>
> |Round|CLIP↑|Color(MS-SWD↓)|Color(C-Hist↓)|Color(GPT4o↑)|Texture(CLIP-I↑)|Texture(KID↓)|
> |:--:|:---:|:--:|:--:|:-:|:----:|:-:|
> |1|0.301|6.08|0.95|3.14|0.751|0.064|
> |2|0.301|6.05|0.95|3.15|0.750|0.065|
> |3|0.300|6.23|0.96|3.12|0.752|0.064|
> |4|0.301|6.16|0.95|3.12|0.751|0.066|
> |5|0.304|6.08|0.95|3.14|0.749|0.066|
>
> **Q5(support for multiple-image control)**:  Our method supports simultaneous control over multiple colors and textures to generate images with mixed color and texture attributes. To ensure a clearer and more direct comparison of evaluation metrics, we use two color reference images and two texture reference images as inputs in each experiment. The following two tables summarize the results: the first table reports *multi-color control* using two color references and one texture reference, while the second table reports *multi-texture control* using two texture references and one color reference. Complex content prompts are sampled from **Q2**.
> For *multi-color control*, the comparison methods use concatenated textual prompts containing two color descriptions (generated by GPT-4o) to achieve multi-color control. Experimental results demonstrate that our method consistently outperforms others in color fidelity while maintaining competitive texture consistency. Notably, although IP-Adapter obtains the highest numerical score for texture consistency, it exhibits severe semantic leakage from the texture reference image (Row 2 and Row 3 in Fig. 5), resulting in poor text-image alignment (CLIP: 0.269), which is insufficient for effective stylization. In contrast, our method achieves a better balance between color and texture consistency, demonstrating superior disentanglement and control over color and texture throughout the stylization process.
>
> |Method|CLIP↑|Color (MS-SWD↓)| Color (C-Hist↓)|Color (GPT4o↑)|Texture (CLIP-I↑)|Texture(KID↓)|
> |:-:|:-:|:-:|:-:|:-:|:-:|:-:|
> |SDXL|0.290|9.06|1.148|2.98|0.695|0.093|
> |IP-Adapter|0.259|10.31|1.185|2.87|0.814|0.060|
> |InstantStyle|0.278|11.23|1.204|2.81|0.749|0.067|
> |Artist|0.262|12.55|1.212|2.40|0.743|0.071|
> |DEADiff|0.275|10.48|1.138|2.66|0.717|0.089|
> |StyleDrop|0.285|11.76|1.173|2.74|0.723|0.077|
> |dreamstyler|0.291|16.54|1.222|2.08|0.679|0.094|
> |CSGO|0.300|12.81|1.213|2.58|0.729|0.081|
> |SADis(Ours)|0.302|7.27|1.002|3.06|0.750|0.066|
>
> For *multi-texture control*, since other methods do not support multiple texture reference images as input, we report results only for our method in this setting. The generated visualizations demonstrate that our approach effectively blends multiple color and texture attributes. In future versions, we will provide additional experimental results on the mixing of multiple colors and textures in the supplementary materials.
>
> |Method|CLIP↑|Color(MS-SWD↓)|Color(C-Hist↓)|Color(GPT4o↑)|Texture(CLIP-I↑)|Texture(KID↓)|
> |:-:|:-:|:-:|:-:|:-:|:-:|:-:|
> |SADis|0.305|6.14|0.955|3.14|0.749|0.071|
>
> **Paper formatting concerns**: Thanks for pointing out our grammatical and logical errors in the original paper, we will correct them in any future versions.
>
> [1] Concept Sliders: LoRA Adaptors for Precise Control in Diffusion Models (ECCV 2024)  \
> [2] IP-Composer: Semantic Composition of Visual Concepts. (SIGGRAPH 2025) \
> [3] SEGA: Instructing Diffusion using Semantic Dimensions (NIPS 2023)  \
> [4] ToMe: Token Merging for Training-Free Semantic Binding in Text-to-Image Synthesis (NIPS 2024) \
> [5] WNNM: Weighted Nuclear Norm Minimization with Application to Image Denoising (CVPR 2014) \
> [6] DreamBench++: A Human-Aligned Benchmark for Personalized Image Generation (ICLR 2025)

---

> > ### Author Response · Authors · 2025-08-07
> >
> > Dear Reviewer L5GK,
> >
> > Thank you so much for your valuable comments. Your suggestions on complex prompts and multi-image control are very helpful. As the limited time remaining for the author-reviewer discussion, we are eager to know whether our responses have adequately addressed your concerns.
> >
> > We sincerely appreciate your thoughtful feedback and we hope that our response has addressed your concerns.
> >
> > Best regards,
> >
> > The Authors

---

> > > ### Comment · Reviewer_L5Gk · 2025-08-07
> > > **Official Comment by Reviewer L5Gk**
> > >
> > > Thanks for the rebuttal. My concerns have been addressed. I recommend including these analyses in the main paper to improve clarity and completeness. Based on this, I am increasing my score.

---

> > > > ### Author Response · Authors · 2025-08-07
> > > >
> > > > Dear Reviewer L5GK,
> > > >
> > > > Thank you very much for your constructive comments and positive feedback. We greatly appreciate your thoughtful suggestions throughout the review process. We will include these analyses in the main paper to further enhance its clarity and completeness.
> > > >
> > > > Best regards,
> > > >
> > > > The Authors

---

### Note · Authors · 2025-08-12

Dear Reviewers and Area Chairs.

We sincerely thank all reviewers (**R1 L5Gk**, **R2 N4y7**, **R3 Z8Hi**, **R4 3c1E**) and **Area Chairs** for their valuable time and insightful comments.

---

We are pleased to note that:
1. **R2 N4y7** and **R4 3c1E** recognize that our method's **performance is strong** and results are with **good quality**.

2. **R1 L5Gk** and **R3 Z8Hi** think our paper is **well-written**,  the **motivation is clear**, and the findings are **interesting**.

3. **R3 Z8Hi** thinks the **evaluation is thorough** and demonstrates the effectiveness of the proposed method.

4. **R2 N4y7** finds that our method is **applicable**, **addresses practical needs**, and the **design is reasonable**.

5. **R4 3c1E** consider that the method is **compatible with the community ecosystem**.

6. According to the reviewers' comments, all of the reviewers agree that we **have addressed their concerns**.

---

We have responded to each reviewer individually and would like to summarize our responses here:

1. We have supplemented experiments to evaluate **stability of generating multiple images (R1 L5Gk)**, **complex prompts (R1 L5Gk)**, and **experiments with the same reference image (R3 Z8Hi)**. According to the reviewers' feedback, the results have addressed their concerns.

2. We have also verified that our method  **supports for multiple-image control (R1 L5Gk)**, and **is compatible with other image encoders (DINOv2 and SigLIP) (R2 N4y7) as well as MMDiT models (SD3.5 and FLUX.1-dev) (R4 3c1E)**. The results demonstrate our method's good extensibility.

3. We have clarified some misunderstandings of reviewers, including **hypothesis of Eq.1 and Eq.3  (R1 L5Gk)**, **texture performance (R2 N4y7 and R4 3c1E)**,  and **the TextureExtraction module (R4 3c1E)**. The reviewers have returned feedback confirming that we have addressed their concerns.

4. We have added **ablation studies on the hyperparameters γ, β and cross-attention layers (R4 3c1E)**.

---

We would like to once again express our sincere gratitude to all reviewers and area chairs. The updated results generated during the rebuttal period will be incorporated into the final version of the paper.

Best regards,

Authors

---

### Decision · Program_Chairs · 2025-09-17

**Decision:**

Accept (poster)

**Comment:**

This paper receives positive ratings of (4, 4, 4, 5). The reviewers acknowledge the simplicity and tuning-free nature of the proposed method, thorough evaluation, and the presentation. Most of the concerns are addressed in the rebuttal. After reading the paper, review, and response, the AC finds no reason to overturn the decision. An acceptance is recommended. The authors should include the discussion in the rebuttal to further improve the paper.